# A population modification gene drive targeting both *Saglin* and *Lipophorin* impairs *Plasmodium* transmission in *Anopheles* mosquitoes

Emily I Green, Etienne Jaouen, Dennis Klug, Roenick Proveti Olmo, Amandine Gautier, Stéphanie Blandin, Eric Marois*

Inserm U1257, CNRS UPR9022, University of Strasbourg, Strasbourg, France

**Abstract** Lipophorin is an essential, highly expressed lipid transport protein that is secreted and circulates in insect hemolymph. We hijacked the *Anopheles coluzzii Lipophorin* gene to make it co-express a single-chain version of antibody 2A10, which binds sporozoites of the malaria parasite *Plasmodium falciparum*. The resulting transgenic mosquitoes show a markedly decreased ability to transmit *Plasmodium berghei* expressing the *P. falciparum* circumsporozoite protein to mice. To force the spread of this antimalarial transgene in a mosquito population, we designed and tested several CRISPR/Cas9-based gene drives. One of these is installed in, and disrupts, the pro-parasitic gene *Saglin* and also cleaves wild-type *Lipophorin,* causing the anti-malarial modified *Lipophorin* version to replace the wild type and hitch-hike together with the *Saglin* drive. Although generating drive-resistant alleles and showing instability in its gRNA-encoding multiplex array, the *Saglin*-based gene drive reached high levels in caged mosquito populations and efficiently promoted the simultaneous spread of the antimalarial *Lipophorin::Sc2A10* allele. This combination is expected to decrease parasite transmission via two different mechanisms. This work contributes to the design of novel strategies to spread antimalarial transgenes in mosquitoes, and illustrates some expected and unexpected outcomes encountered when establishing a population modification gene drive.

## Editor's evaluation

This study presents the generation of a two-component gene drive for population modification in Anopheles coluzzii, a major malaria vector in Africa. By testing multiple elegant drive designs, the authors convincingly achieve the spread of antimalarial cargos in caged mosquito populations. Overall, this work represents a significant advance towards a possible application of genetic technologies for malaria control.

## Introduction

Malaria-transmitting mosquitoes cause over 245 million malaria cases and about 620,000 deaths annually (***WHO, 2021***). Vector control, mainly in the form of insecticide indoor residual spraying and insecticide-treated bed nets, has permitted a steady decrease in malaria cases during the last decades. However, this decline is stalling as mosquitoes become increasingly resistant to insecticides and as malaria parasites continue to evolve resistance to drugs. These ongoing genetic changes in vectors and parasites, under the selection pressure of two major pillars of malaria control, jeopardize the hard-won successes in the fight against malaria. To address the need for complementary malaria control approaches, intervention strategies based on intentional genetic changes in mosquitoes to

*For correspondence:
e.marois@unistra.fr

**Competing interest:** The authors declare that no competing interests exist.

decrease malaria transmission are under development. Recent advances in the field of gene drive (GD) provide a mean to push desirable transgenes to invade a target population of insects (*Carballar-Lejarazú and James, 2017*; *Gantz and Akbari, 2018*; *Quinn and Nolan, 2020*; *James and Santos, 2023*).

GD interventions against malaria mosquitoes are still in the laboratory phase of their development, and can be subdivided in two classes: (a) population suppression GDs that spread either female sterility or female killing in mosquito populations with the aim to eliminate them (*Hammond et al., 2016*; *Kyrou et al., 2018*; *Simoni et al., 2020*); (b) population modification GDs that leave mosquito populations in place, but genetically alter them to reduce their disease transmission capacity (*Gantz et al., 2015*; *Pham et al., 2019*; *Carballar-Lejarazú et al., 2020*; *Carballar-Lejarazú et al., 2023*). Both approaches raise ethical, safety and ecological concerns, and trigger controversy (e.g., *National Academies of Science, Engineering and Medicine, 2016*; *Courtier-Orgogozo et al., 2017*; *de Graeff et al., 2021*). One concern raised by elimination approaches is their possible consequences on the food web and ecosystems, even where dozens of sympatric mosquito species are potentially available to compensate for the loss of the target species. In some regions, elimination of one species could favor the installation of competing species that could also act as vectors, as can be feared for the *Aedes aegypti* / *Ae. albopictus* mosquito dyad vectoring dengue, Zika, and chikungunya viruses. Modification approaches alleviate these concerns by leaving the target species in its ecological niche, but in turn, the indefinite persistence of transgenes utilized in modification drives could have long-term secondary effects, so that both approaches should be planned with utmost caution. However, the option of rejecting gene drive technologies as a complementary tool to fight mosquito-borne pathogens also raises ethical issues, by ignoring a chance to curb the immense mortality and morbidity caused by mosquito-borne infections.

*Anopheles gambiae* (sensu lato), the major mosquito vector of malaria in Sub-Saharan Africa, is widely distributed, locally abundant during rainy seasons and its biomass might significantly contribute to the food chain, in a temporal fashion. Besides their possible negative impact on food webs, elimination approaches for this mosquito are likely to be complicated by re-colonization from residual pockets of surviving isolated populations, resulting in chasing dynamics of gene drive (*Champer et al., 2021*; *Champer et al., 2022*). For this species in its native ecosystem, population modification to decrease vector competence and preserve ecological functions could be a meaningful gene drive approach. Here, we present the engineering of the *Lipophorin* (*Lp*) essential gene in *Anopheles coluzzii*, a prominent member of the *A. gambiae* species complex and a major malaria vector in sub-Saharan Africa.

*Lp* naturally encodes an abundant lipid transport protein secreted in mosquito hemolymph. We modified *Lp* to co-express a single-chain antibody (ScFv) derived from 2A10, a well-established mouse monoclonal antibody that binds the *Plasmodium falciparum* circumsporozoite protein (CSP) (*Zavala et al., 1983*). Sporozoites are the parasite stage that mosquito females inject in vertebrate hosts while aquiring a blood meal. Once in the host, sporozoites quickly infect the liver and multiply to produce merozoites with a tropism for red blood cells. Through its binding to CSP (the major sporozoite surface antigen), 2A10 was shown to decrease sporozoite infectivity to cultured human hepatocytes (*Hollingdale et al., 1984*). Transgenic expression of a 2A10 ScFv, fused to the antimicrobial peptide Cecropin A and expressed under the control of the *Vitellogenin* promoter, decreased the number of salivary gland sporozoites in *Anopheles stephensi* (*Isaacs et al., 2011*; *Isaacs et al., 2012*) and in *A. gambiae*/*coluzzii* (*Carballar-Lejarazú et al., 2023*). Expression of a 2A10 ScFv directly in the salivary glands of *An. stephensi* reduced sporozoite infectivity to cultured hepatocytes, and transmission to mice of chimeric *Plasmodium berghei* expressing CSP from *P. falciparum* (*Sumitani et al., 2013*). Likewise, expression of a different CSP-binding ScFv in transgenic *A. coluzzii* decreased the sporozoite's ability to infect cultured hepatocytes and to cause malaria in mice (*Triller et al., 2017*).

We inserted the 2A10 ScFv coding sequence within the endogenous open-reading frame of the essential *Lp* gene, preserving Lp function and exploiting a natural proteolytic cleavage site to separate the fusion protein moieties during their secretion. Using a mouse model of malaria, in which a transgenic rodent *Plasmodium berghei* strain expresses CSP from *P. falciparum*, we show a strong decrease in the ability of the genetically engineered mosquitoes to transmit *Plasmodium*. In a second step, we tested different CRISPR/Cas9-based GD designs to force the spread of the *Lp::Sc2A10* transgene in laboratory mosquito populations. A first, suppression GD disrupting the wild-type (WT) *Lp* locus revealed haploinsufficiency of the *Lp* gene, precluding this approach. Other GD versions were

designed to home into and disrupt the *Saglin* pro-parasitic gene on the X chromosome. These GD also expressed guide RNAs targeting WT *Lp*, promoting homing of *Lp::Sc2A10* on the second chromosome. This dual homing at both the *Saglin* and *Lp* loci resulted in the simultaneous spread of the two transgenes in caged mosquito populations. In modified mosquitoes, sporozoite transmission should be reduced by two distinct mechanisms, i.e., depriving *Plasmodium* from its Saglin agonist plus attacking sporozoites with the ScFv. Each of the *Saglin* and the *Lp* loci were targeted with multiplexes of three gRNAs, to increase the chance that failed homing events result in loss-of-function *Lp* mutations that would be eliminated by natural selection, and loss-of-function, GD-resistant *Saglin* alleles that should still contribute to decrease vector competence. One homing-refractory functional allele was however observed at the *Lp* locus.

## Results

### Development of an engineered Lp allele encoding an anti-sporozoite factor while preserving Lp function

The two most abundant proteins secreted in *Anopheles* hemolymph are the nutrient transporters Lipophorin (Lp) and Vitellogenin (Vg), as illustrated by Coomassie staining of electrophoresis gels revealing both proteins as the major visible bands in hemolymph extracts (*Figure 1A* and *Rono et al., 2010*). While Vg is induced in mosquito females only upon blood feeding and wanes within 3 days as eggs develop, Lp is expressed constitutively, with a further spike of expression after blood feeding (*Attardo et al., 2005*; *Rono et al., 2010*). We reasoned that Lp protein abundance, secretion in hemolymph and persistence of expression outside gonotrophic cycles make *Lp* a particularly attractive host gene to co-express and secrete high levels of synthetic anti-*Plasmodium* factors. Here we tested a 2A10 scFv version without fusion to Cecropin as antimalarial effector, and hereinafter refer to this effector as Sc2A10. Using the CRISPR-Cas9 system, we knocked-in codon-optimized *Sc2A10* in frame after the endogenous first *Lp* coding exon, which encodes the Lp secretion signal peptide (*Figure 1B*, for sequence of the plasmid used for knock-in see *Supplementary file 1A*). As a result, transcription of the chimeric *Lp::Sc2A10* mRNA encodes the natural Lp signal peptide followed by Sc2A10 fused to the remainder of the Lp protein sequence. A natural furin proteolytic cleavage site located between the ApoLpII and ApoLpI subunits of the Lp polypeptide allows their separation during maturation in the Golgi apparatus (*Smolenaars et al., 2005*). We duplicated this motif to detach Sc2A10 from ApoLpII during protein maturation. The knock-in design also introduced silent mutations in the majority of possible Cas9 sgRNA target sites within *Lp* exon 1 and the beginning of intron 1, to facilitate selection of multiple sgRNAs targeting WT *Lp*, but not its engineered allele, when designing subsequent GDs. Finally, the 1644-bp first intron immediately following *Lp* exon 1 offered an ideal location to host a fluorescent selection marker to identify and track the modified gene. We placed a *GFP* marker gene under control of the synthetic *3xP3* promoter (*Horn and Wimmer, 2000*) and *D. melanogaster Tubulin56D* terminator in this intron, with care taken to preserve the intron splice junctions and in reverse orientation relative to *Lp* transcription so that the *Tub56D* terminator would not cause premature arrest of *Lp* mRNA transcription. The *Lp* promoter is active in the fat body, while the *3xP3* promoter is active in the nervous system. These synthetic modifications in the *Lp* locus are schematized in *Figure 1B* and annotated in the provided sequence file (*Supplementary file 1B*).

Using automated flow cytometry-based (COPAS) selection of live neonate larvae, based on differential GFP expression between homozygotes and heterozygotes (*Marois et al., 2012*), we established two stocks of *Lp::Sc2A10* mosquitoes, one heterozygous, the other homozygous. The homozygous stock was fertile and viable, devoid of obvious fitness cost. This indicates that the transgenic insertion into *Lp* did not disrupt its essential functions in development, physiology, flight, or reproduction. A more careful evaluation of *Lp::Sc2A10* fitness costs was conducted by following the dynamics of the transgene in the heterozygous population over 17 generations. This revealed slow disappearance of the transgene over time (*Figure 1C* and *Supplementary file 2*) indicative of a subtle fitness cost that decreased transgene frequency by an average of 2.3% each generation. In fertility assays comparing WT and homozygous transgenic females, the latter consistently engendered fewer progeny than their WT sisters, pointing to a negative impact of the *Lp* modification on their reproductive capacity (*Supplementary file 3*).

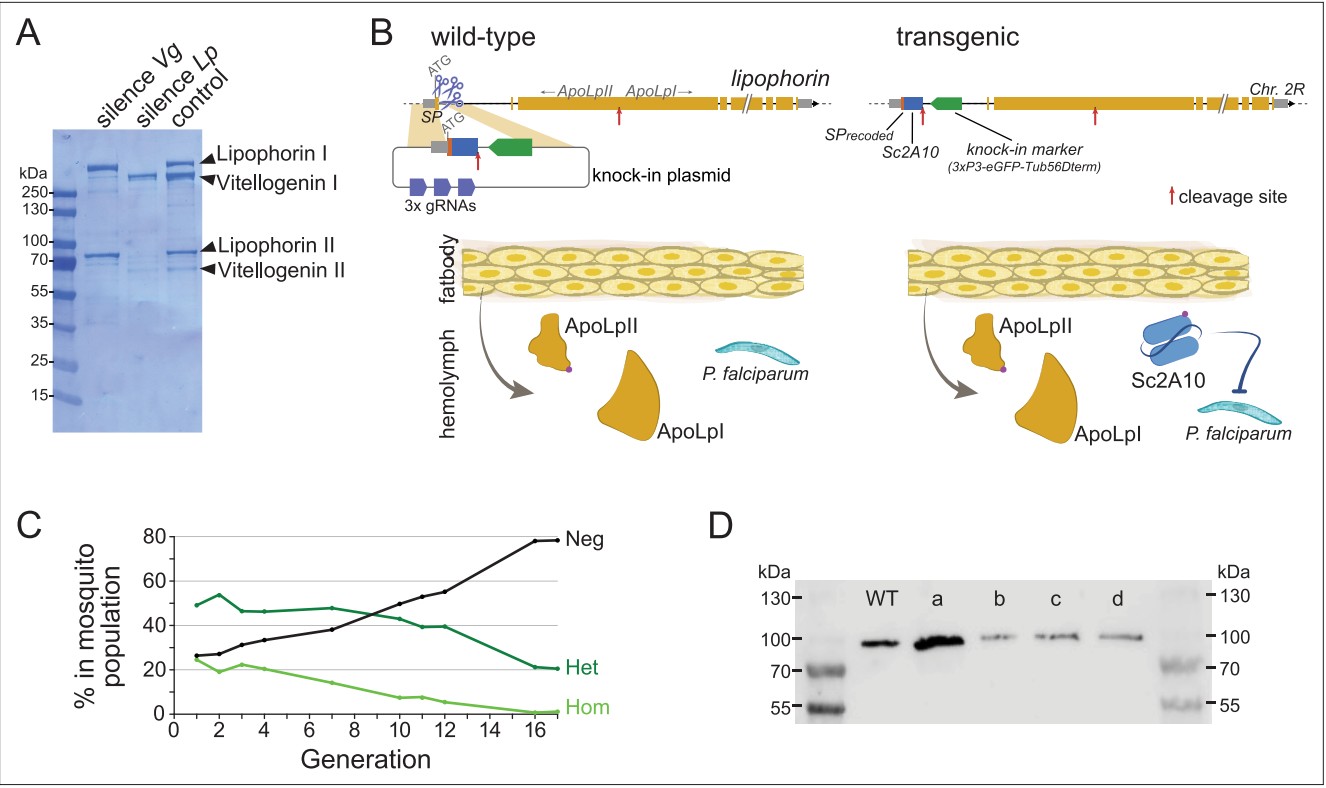

**Figure 1.** Characterization and modification of the endogenous *Lipophorin* gene. (**A**) Coomassie staining of electrophoresed hemolymph proteins. Mosquitoes were injected with the indicated double-stranded RNA to silence either *Vg* or *Lp* as described (**Rono et al., 2010**) and offered a blood meal after 2 days. Hemolymph was collected 42 hr post blood feeding. (**B**) Scheme (not to scale) depicting the insertion of synthetic sequences into the *Lp* gene and secretion of the Sc2A10 single chain antibody from fat body cells. The Signal Peptide (SP) was recoded to remove gRNA target sites. Sc2A10 immediately follows the SP and ends with a RFRR furin cleavage site (red arrow) for separation from ApoLpII in the secretory pathway. This site is duplicated from the ApoLpI/II natural furin cleavage site found downstream. Fat body cells of mosquitoes bearing the *Lp::Sc2A10* transgene are thus expected to constitutively secrete Sc2A10, ApoLpII and ApoLpI in the hemolymph. (**C**) Spontaneous dynamics of *Lp::Sc2A10* transgene frequency over 17 generations. Between 1200 and 9100 neonate larvae in generations 1, 2, 3, 4, 7, 10, 11, 12, and 17, and 522 larvae in generation 16, were analysed by COPAS flow cytometry. The proportions of homozygotes (Hom, highest eGFP intensity), heterozygotes (Het, intermediate eGFP intensity) and eGFP negative (Neg) were estimated for each analyzed generation by gating each larval population in the COPAS software. Counts are provided in *Supplementary file 2*. (**D**) Western blot with anti-ApoLpII antibody on hemolymph samples from 10 female mosquitoes. First and last lanes: protein size ladder (molecular weights indicated), second lane: hemolymph from WT mosquitoes. Lanes c, d: hemolymph from homozygous *Lp::Sc2A10* mosquitoes. Failed cleavage between scFv and ApoLpII would result in a 13 kDa molecular weight upshift compared to the control. Lanes a, b: hemolymph from mosquitoes expressing a distinct *Lp::ScFv* fusion not further discussed in this work.

The online version of this article includes the following source data for figure 1:

**Source data 1.** Coomassie staining of an SDS-PAGE gel with electrophoresed hemolymph proteins.

**Source data 2.** Coomassie staining of an SDS-PAGE gel with electrophoresed hemolymph proteins.

**Source data 3.** Western blot with anti-ApoLpII antibody on hemolymph samples from 10 female mosquitoes.

**Source data 4.** Western blot with anti-ApoLpII antibody on hemolymph samples from 10 female mosquitoes.

## The Lp::Sc2A10 transgene directs Sc2A10 secretion into mosquito hemolymph

To test whether the Sc2A10 single-chain antibody was expressed and secreted in hemolymph via the Lp signal peptide, we extracted hemolymph from transgenic female mosquitoes and analyzed its protein content by mass spectrometry. Because the *Lp::Sc2A10* gene fusion produces a single mRNA, with the endogenous Lp signal peptide addressing the entire fusion protein to the secretory pathway, the Sc2A10 protein fragment must initially be stoichiometric to the Lp subunits ApoLpI and ApoLpII. Upon proteolytic cleavage dissociating Sc2A10, ApoLpII, and ApoLpI and after their secretion into the hemolymph, each moiety possibly follows a different fate. In two mass spectrometry samples from the hemolymph of homozygous transgenic mosquitoes, for a total of 2629 peptides from the largest

**Table 1.** Mass spectrometry identification of peptides from Lipophorin and Sc2A10 in hemolymph from transgenic mosquitoes and wild-type sibling controls.

The table shows the total number of peptide spectra detected in each hemolymph sample for the two Lp subunits and for Sc2A10. Hemolymph was collected from homozygous (hom) and heterozygous (het) transgenic mosquitoes. Relative, normalized abundance of these peptides in the protein sample, calculated by dividing their spectral count by the respective protein's molecular weight (kDa) and by the total spectral counts from known abundant hemolymph proteins found in the sample (ApoLpIII, APL1C, LRIM1, Nimrod, Phenoloxidase, and TEP1), is indicated in parentheses. Resulting values were multiplied by a constant factor equal to the average number of spectra from these control proteins across samples.

| Sample | ApoLpI | ApoLpII | Sc2A10 |
|---|---|---|---|
| WT control | 2075 (5.8) | 486 (5.1) | 0 (0) |
| hom Sc2A10, sample 1 | 1627 (5.1) | 405 (4.8) | 11 (0.4) |
| hom Sc2A10, sample 2 | 1002 (4.4) | 267 (4.4) | 7 (0.3) |
| het Sc2A10, sample 1 | 1557 (10.1) | 351 (8.7) | 2 (0.1) |
| het Sc2A10, sample 2 | 2290 (6.0) | 664 (6.7) | 2 (0.1) |

Lp subunit ApoLpI, we detected 672 peptides from smaller ApoLpII and 18 from Sc2A10 (*Table 1*). Corrected for size, this suggests that Sc2A10 molecules are 10- to 20-fold less abundant in the hemolymph compared to ApoLpI and ApoLpII. This observation suggests that while Sc2A10 is made and secreted into hemolymph, its disappearance by degradation, uptake in cells or stickiness to tissue, is faster than that of its sister Lp proteins. Of note, western blots using anti-ApoLpII antibodies on hemolymph samples from homozygous transgenic mosquitoes showed no size upshift for ApoLpII, confirming that Sc2A10 is properly released from ApoLpII (*Figure 1D*). Overall, we observed higher Lp concentrations in heterozygous compared to homozygous mosquitoes, and a three to fourfold lower Sc2A10: Lp ratio in heterozygotes compared to homozygotes, indicating that the chimeric proteins from *Lp::Sc2A10* are less expressed than from WT *Lp*.

## The Lp::Sc2A10 transgene decreases *Plasmodium* transmission

To assess whether the amount of Sc2A10 present in hemolymph can affect *Plasmodium* transmission, we infected homozygous transgenic mosquitoes and their control wild-type siblings and allowed parasites to develop for 16–20 days, before exposing naïve mice to be bitten by groups of 10 infected mosquitoes per mouse. The *Plasmodium berghei* strain we employed, *Pb-PfCSP*hsp70-GFP, expresses GFP at a high level under control of the constitutive *hsp70* promoter and has its native CSP substituted with that from *P. falciparum* (*Manzoni et al., 2014*; *Triller et al., 2017* and see Materials and methods), which is the antigen recognized by 2A10. Only mosquitoes displaying GFP parasites visible through the cuticle were used to infect mice. We controlled that each mouse was bitten by at least six infected mosquitoes, with exceptions indicated in *Supplementary file 4*, by counting the blood-engorged females.

Consistent with previous reports (*Isaacs et al., 2011*; *Isaacs et al., 2012*; *Sumitani et al., 2013*), Sc2A10 significantly reduced transmission of *Pb-PfCSP* (p<0.0001). Only 29.7% (11/37) of mice exposed to *Lp::Sc2A10* homozygous infected mosquitoes developed parasitemia, compared to 97.1% (33/34) exposed to infected control mosquitoes (*Figure 2A* and *Supplementary file 4A*). Additionally, 6 of the 11 mice that did become infected by transgenic mosquitoes experienced a 1 day delay in the development of detectable parasitemia in the blood (*Supplementary file 4A*), indicative of reduced liver infection (*Reuling et al., 2020*).

We examined if the transmission-reducing activity of the *Lp::Sc2A10* transgene is dependent on the dosage of Sc2A10, that is whether a single copy of the transgene would be as efficient as two. For this we exposed naive mice to groups of 10 infected mosquitoes carrying zero, one or two *Lp::Sc2A10* copies. While 17 out of 18 mice bitten by control mosquitoes became infected, only 4 out of 12 mice bitten by heterozygous mosquitoes became infected (*Figure 2B* and *Supplementary file 4B*). Two-tailed Fisher's exact test shows a statistically significant reduction when comparing wild-type

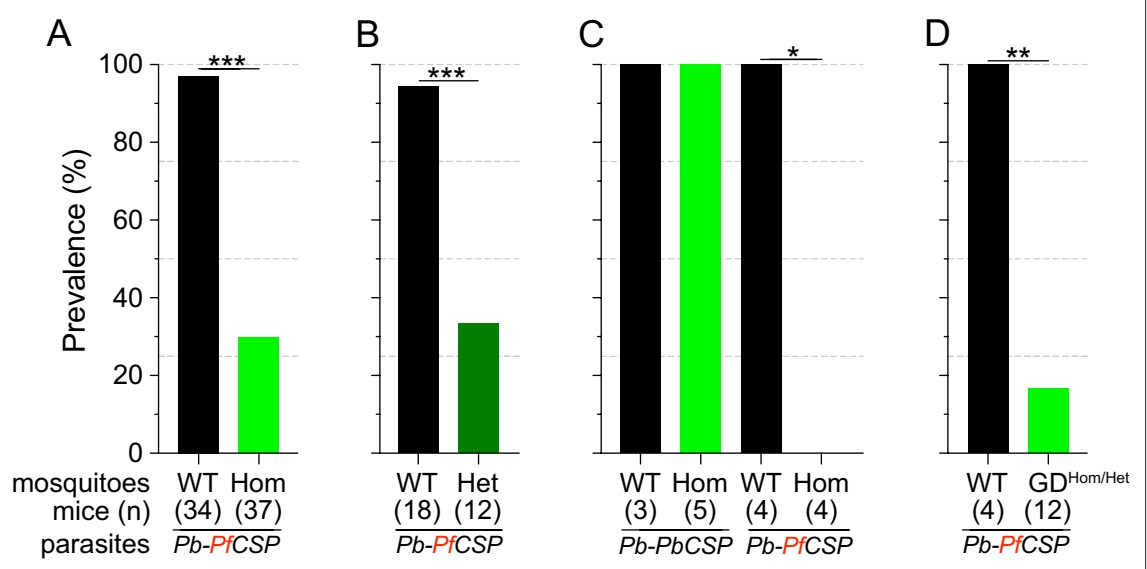

**Figure 2.** Infection status of mice bitten by Sc2A10-expressing vs. WT mosquitoes infected with *Plasmodium*. Mosquito genotype and number of mice exposed to infectious mosquito bites is indicated below the bar for each condition. (**A**) Comparison of the percentages of infected mice after exposure to the bites of homozygous transgenic *Lp::Sc2A10* mosquito females or their wild-type siblings, infected with a *Plasmodium berghei* strain expressing *P. falciparum* CSP. Aggregated data from 11 independent experiments performed over a period of two years were analyzed using Fisher's exact test with a two-tailed p-value (p<0.0001). (**B**) Comparison of mouse infection upon exposure to heterozygous transgenic *Lp::Sc2A10* mosquito females or their wild-type siblings, aggregated data from 5 independent experiments (p=0.0006). (**C**) Mice exposed to WT or homozygous *Lp::Sc2A10* mosquitoes infected with either *P. berghei* expressing *P. falciparum* CSP or *P. berghei* expressing endogenous *P. berghei* CSP (single experiment, p=0.0286). (**D**) Mice exposed to WT or to *SagGD^vasa* Gene Drive and *Lp::Sc2A10* carrying mosquitoes, infected with a *Plasmodium berghei* strain expressing *P. falciparum* CSP (single experiment, p=0.0082).

and heterozygous (p=0.0006) mosquitoes regarding their *Plasmodium*-transmitting capacity. Therefore, heterozygous mosquitoes showed a transmission blocking activity comparable to that seen in homozygotes.

To verify that the transmission-blocking phenotype of the *Lp::Sc2A10* transgene was specific to CSP from *P. falciparum*, we performed a control bite-back experiment comparing *Pb-PfCSP*^hsp70-GFP to its *Pb-hsp70-GFP* parental parasite strain expressing endogenous *PbCSP*. While *Lp::Sc2A10* and wild-type mosquitoes transmitted the control *P. berghei* strain to mice with equal efficiencies (n=5/5 and 4/4 mice infected, respectively), only the WT mosquitoes transmitted *Pb-PfCSP*^hsp70-GFP (n=3/3 mice infected), whereas *Lp::Sc2A10* mosquitoes blocked transmission of the *Pf*CSP-expressing strain (n=0/4 mice infected). This indicates that the transmission blocking capacity of the *Sc2A10* transgene is indeed *Pf*CSP-specific (**Figure 2C** and **Supplementary file 4C**).

## Testing a suppression gene drive homing into wild-type Lp to push Lp::Sc2A10 toward fixation

To push the *Lp::Sc2A10* anti-*Plasmodium* gene towards fixation in a mosquito population, our first strategy was to employ a transiently acting Cas9-based GD destroying the wild-type *Lp* gene (**Figure 3A**). This aimed to exploit the natural tendency of suppression GDs that target an essential gene to initially spread in the population only to be supplanted, over time, by GD-immune functional mutants of the target gene (**Hammond et al., 2017**; **Champer et al., 2018**; **Carballar-Lejarazú et al., 2022**). Here *Lp::Sc2A10* itself, lacking the gRNA target sites, would play the role of a GD-immune, functional *Lp* mutant allele. The advantage of this approach was that a *Cas9* and gRNA-carrying transgenic GD cassette inserted in *Lp* need not persist indefinitely in the population, being supplanted by the technologically more benign *Lp::Sc2A10* transgene. To minimize the spontaneous emergence of undesirable additional GD-immune and functional *Lp* alleles, we used a multiplex of 4 gRNAs targeting *Lp*. The rationale was that in cases of failed GD homing, multiple Cas9 cuts repaired by non-homologous end joining (NHEJ) would mainly generate loss-of-function alleles in the essential *Lp* gene (e.g. deletions between Cas9 cut sites, frameshifts at several individual cut sites) doomed to be

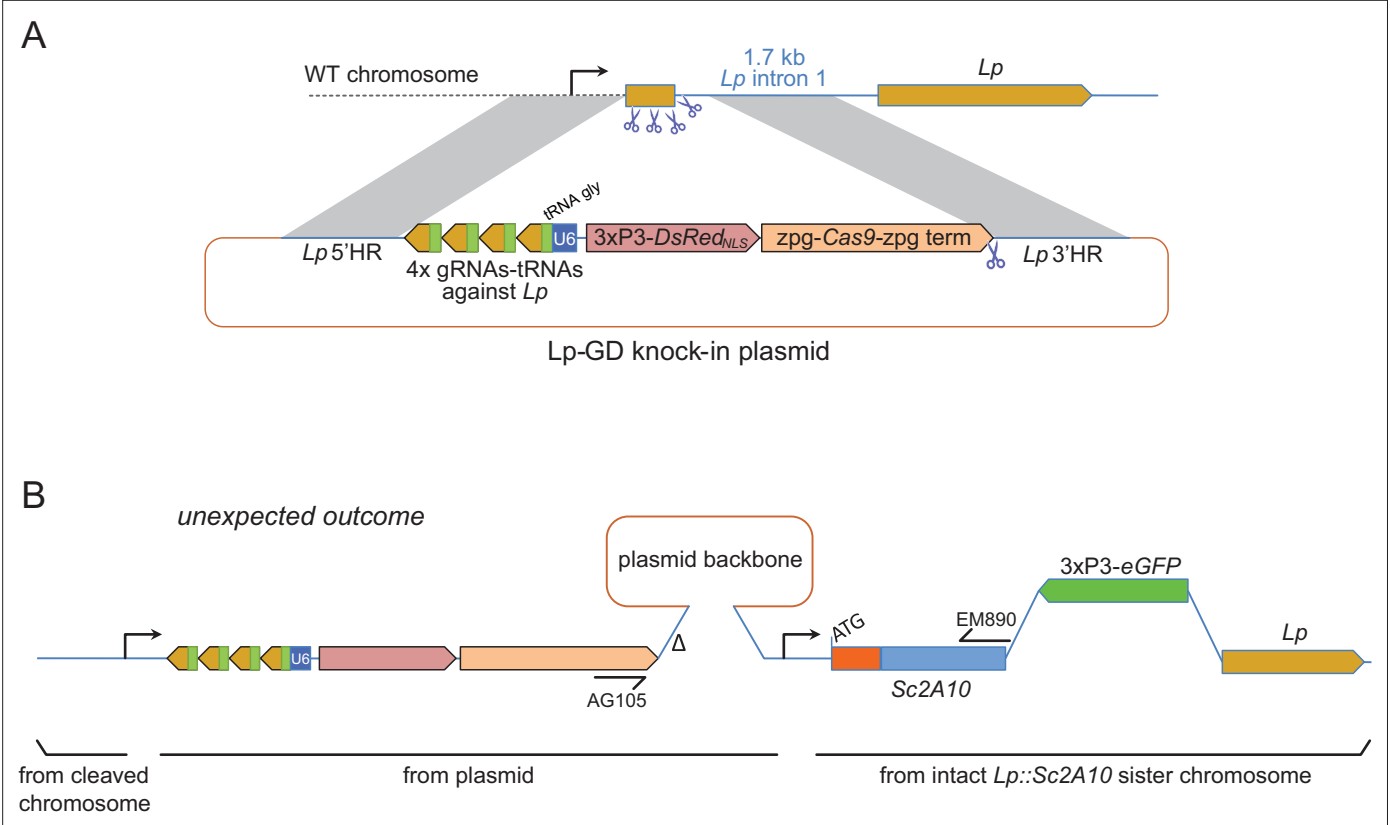

**Figure 3.** Insertion of a *Lipophorin* suppression gene drive and its expected and unexpected outcomes. (**A**) Scheme of the suppression drive construct. Upon chromosome cleavage by Cas9 (shown by scissors), homologous recombination (gray shading) via 5′ and 3′ regions of homology (HR) is expected to insert the intervening elements (gRNA-encoding array, zpg-Cas9 expression cassette, DsRed$_{NLS}$ marker cassette) into the beginning of the Lp gene, disrupting its function. Note that this recombination event knocking out the essential *Lp* gene was designed to occur in the presence of an intact *Lp::Sc2A10* sister chromosome, not shown in the scheme but that contributed to the unexpected outcome shown in B. (**B**) Most knock-in events unexpectedly arose from compound recombinations between the cleaved WT chromosome and both the repair template plasmid and the intact *Lp::Sc2A10* sister chromosome, resulting in insertion of the gene drive elements genetically linked to the *Lp::Sc2A10* transgene. A hypothetical scheme interpreting the compound insertion is shown. It was inferred from the genetic linkage observed between GFP and DsRed$_{NLS}$, and from sequencing a PCR product obtained with primers AG105 and EM890 (indicated). Δ indicates a 500 bp deletion in the cloned Lp 3′region, which arose following micro-injection of the plasmid because one of the gRNA target sites was retained by error in this part of the construct (indicated by scissors). Presence of the gRNA-tRNA array (green boxes +yellow arrows) was confirmed by drive activity in presence of a *vasa-Cas9* transgene.

eliminated by natural selection. We constructed a DsRed$_{NLS}$-marked *Lp* suppression GD (*Figure 3A* and annotated sequence in *Supplementary file 1C*), in which an array of 4 gRNAs separated by tRNAs (*Xie et al., 2015*; *Port and Bullock, 2016*), targeting *Lp* but not *Lp::Sc2A10*, were expressed under control of a single U6 promoter. *Cas9* was placed under the control of *zpg* regulatory elements, currently considered best for use in suppression GDs due to tighter restriction of Cas9 activity to the germ line (*Kyrou et al., 2018*; *Hammond et al., 2021*). Knocking-in this GD into the *Lp* locus was achieved by injecting this plasmid into >800 mosquito eggs carrying a heterozygous copy of GFP-marked *Lp::Sc2A10*, in order to retain a protected functional copy of *Lp* during the disruption of wild-type *Lp* by Cas9. Genomic insertion in injected embryos relied on expression of the construct's own CRISPR/Cas9 components supplemented with an additional heterozygous copy of a YFP-marked *vasa-Cas9* transgene in the embryos.

By screening for the presence of red fluorescence in the G1 progeny of about 140 surviving injected G0 mosquitoes, we recovered several dozens of knock-in G1 larvae (most of which were DsRed$_{NLS}$ and GFP positive) and examined transgene inheritance by crossing them to wild-type mosquitoes. In this cross, DsRed$_{NLS}$ marking the *Lp-GD* construct was expected to strictly segregate away from GFP marking the *Lp::Sc2A10* allele. Puzzlingly however, the majority of knock-in larvae contained a compound *Lp* locus harboring the DsRed$_{NLS}$-marked GD cassette strictly linked to the GFP-marked

*Lp::Sc2A10* transgene. Sequencing PCR products spanning parts of this insertion confirmed that, instead of the expected insertion of the *Lp-GD* cassette, a complex chromosomal rearrangement (*Figure 3B*) had repaired the Cas9-cleaved wild-type chromosome by incorporating the entire injected plasmid, and by recombining with the intact sister chromosome bearing the *Lp::Sc2A10* transgene. The resulting unintended transgenic modification at the *Lp* locus harbored all the components of a classical payload-bearing GD that we hereinafter refer to as *GFP-RFP*. Similarly to the *Lp::Sc2A10* allele, the *GFP-RFP Lp* allele was homozygous viable, functional Lp protein being expressed from its *Lp::Sc2A10* moiety. In spite of the presence of the *zpg-Cas9* and *gRNA*-encoding cassettes in the *GFP-RFP* allele, it was inherited in about 50% of male or female progenies, demonstrating little homing activity of the *GFP-RFP* locus after crosses to WT, except for the appearance of rare GFP-only or RFP-only progeny larvae, raising concern about the efficiency of the *zpg-Cas9* and/or the gRNA-expressing cassettes. Providing an independent copy of *vasa-Cas9* on a different chromosome rescued this defect and resulted in high homing rates of various segments from the *GFP-RFP* cassette (mostly GFP alone, more rarely DsRed$_{NLS}$ alone or both) (*Figure 4A*), indicating that at least some of the gRNAs in the tRNA-spaced array were active and that the defect was in *zpg-Cas9* expression.

Fortunately, we also recovered a minority of knock-in larvae that had integrated the *Lp-GD* construct as expected, marked with DsRed$_{NLS}$ only (*Figure 3A*). However, no evidence of gene drive was observed in further generations, as DsRed inheritance did not exceed 50% when crossing hetero-zygotes to WT. On the contrary, DsRed frequency dropped rapidly in F2 and F3 generations, indicating that the non-functional GD cassette inserted in *Lp* conferred a strong fitness cost. Most heterozygous mosquitoes carrying one copy of *Lp-GD* and either *GFP-RFP* or *Lp::Sc2A10* (both of which should be refractory to homing of the GD construct) showed a striking developmental delay and died before adulthood (*Figure 4B*). Heterozygous mosquitoes carrying *Lp-GD* / WT survived to adulthood but were small and short-lived. These observations indicate that WT *Lp* is largely haplo-insufficient in presence of a *Lp* loss-of-function allele, even more so if WT *Lp* is replaced by *Lp::Sc2A10*. This last point is consistent with the observed fitness cost and lower *Lp* expression from the *Lp::Sc2A10* allele, that already suggested that albeit functional, this allele is inferior to the wild-type.

From the haplo-insufficiency of *Lp* we concluded that it was impossible to create a loss-of-function GD in this locus, precluding an indirect same-locus GD approach. In addition, the *zpg* promoter appeared to have low activity when integrated in the *Lp* locus.

## Driving Lp::Sc2A10 distantly from *Saglin* with a zpg-Cas9 gene drive

To host a GD promoting the spread of *Lp::Sc2A10*, we turned our attention to the mosquito *Saglin* locus, which encodes a *Plasmodium* agonist (*Okulate et al., 2007*; *Ghosh et al., 2009*; *O'Brochta et al., 2019*; *Klug et al., 2023*). Indeed, while not leading to complete *Plasmodium* transmission blockage, the loss of *Saglin* reduces parasite prevalence and loads, reduces transmission, is homozygous viable and has no obvious fitness cost at least in laboratory conditions (*Klug et al., 2023*). Similar to an antimicrobial combination therapy, combining Sc2A10 expression with the knockout of *Saglin* can be expected to reinforce the transmission-blocking properties of each strategy taken alone. Therefore, we sought to disrupt *Saglin* with a gene drive cassette that would home into wild-type *Saglin* and concomitantly promote *Lp::Sc2A10* homing into the *Lp* locus (*Figure 5A*).

The *zpg* promoter still seemed an attractive choice to build a GD dually targeting *Saglin* and *Lp*, to avoid drawbacks from other characterized germline promoters (high maternal Cas9 deposition resulting in failed homing events in the zygote accompanied by the formation of GD refractory alleles; possible somatic mosaic loss-of-function of wild-type *Lp* leading to haplo-insufficiency in heterozygous [*Lp::Sc2A10* /+] mosquitoes). To exclude that the low *zpg-Cas9* activity we previously observed at the *Lp* locus was due to overlooked mutations in our construct, we subcloned the *zpg-Cas9* cassette from a validated GD construct (*Kyrou et al., 2018*) and incorporated it in a *Saglin* knock-in plasmid expressing Cas9, DsRed$_{NLS}$, 3 gRNAs directed against *Saglin* and 4 gRNAs against *Lp* (*Figure 5*, and annotated sequence of this plasmid in *Supplementary file 1D*). Multiplexing the gRNAs was intended to promote the formation of loss-of-function alleles in case of failed homing at the *Lp* and *Saglin* loci: non-functional alleles of the essential *Lp* gene would be eliminated by natural selection, while non-functional *Saglin* alleles would reduce vector competence. From injection of this plasmid into heterozygous *Lp::Sc2A10*/WT embryos, we crossed 36 G0 male survivors to WT, and recovered more than 60 transgenic G1 individuals. The sole source of Cas9 in this injection was the

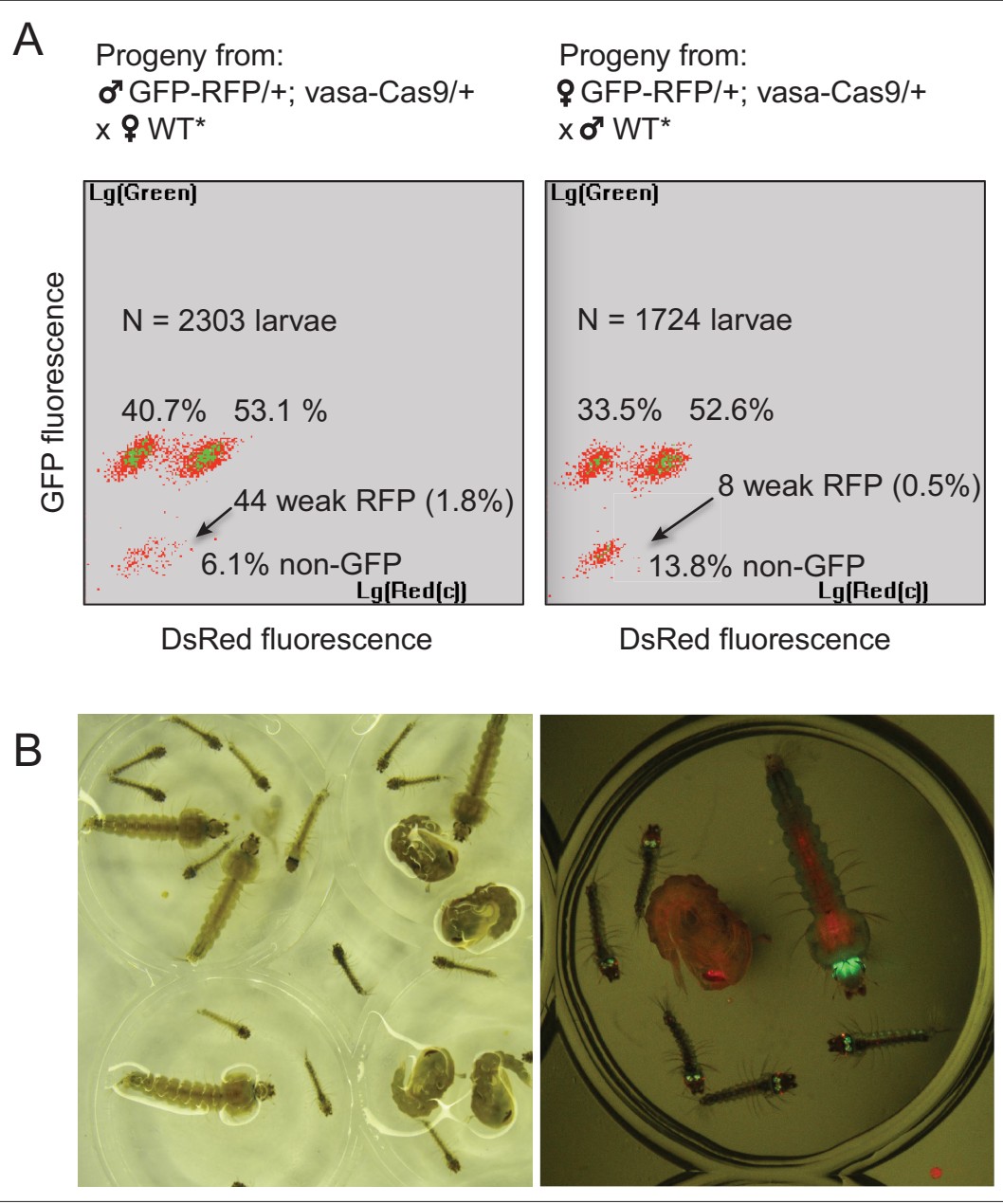

**Figure 4.** Characterization of the two types of Lp-GD integration events. (**A**) The GFP-RFP line showed high homing rates when complemented with an independent vasa-Cas9 transgene. COPAS diagrams show the fluorescence of progeny neonate larvae from heterozygous GFP-RFP mosquitoes (third chromosome) that also carried a non-fluorescent, puromycin resistance-marked vasa-Cas9 transgene (second chromosome, heterozygous), crossed to non-fluorescent partners (WT* indicates non-fluorescent partners actually carrying one copy of the puromycin resistant vasa-Cas9 transgene, not influencing the crossing outcome). N indicates the total number of larvae analyzed in the diagrams; percentages of larvae of each fluorescence are indicated. Non-fluorescent progeny would amount to 50% in the absence of homing. Note that the percentage of GFP-RFP is close to 50%, mainly reflecting Mendelian inheritance of the parental transgene. Thus most homing events involved only a GFP-containing segment from the parental GFP-RFP transgene. Rarer instances of RFP-only segment homing (arrows) were accompanied by a decrease of DsRed fluorescence intensity, indicating that the *3xP3* promoter is less active than in the GFP-RFP context. Total homing amounts to ((40.7+53.1 + 1.8)–50) x2=91.2% in males, ((33.5+52.6 + 0.5)–50) x2=82.2% in females. (**B**) Haploinsufficiency at the *Lipophorin* locus. Photograph shows larval and pupal progeny from heterozygous *Lp-GD* crossed to heterozygous *Lp::Sc2A10*. Right panel focuses on a subset of larvae observed under fluorescent light. Larvae that inherited one copy of the *Lp-GD* loss-of-function allele (red eyes) and

*Figure 4 continued on next page*

*Figure 4 continued*

one copy of *Lp::Sc2A10* (green eyes/brain) are developmentally delayed and will die before adulthood. Individuals that inherited a WT *Lp* copy and either modified allele will complete development, with strong loss of fitness for WT/*Lp-GD* individuals (red-eyed pupa).

knock-in GD plasmid itself, showing that *zpg-Cas9* was efficiently active at least for initial insertion. 100% of the G1 transgenics arising from injected males were females, an indication that the construct had correctly homed onto the X chromosome, where *Saglin* is located. In addition, all G1 transgenics had also inherited a copy of *Lp::Sc2A10* (GFP positive), instead of the expected 50%. This suggests that the GD construct acted on the *Lp* locus, causing either efficient *Lp::Sc2A10* homing already in the germ line of G0 males, and/or the death of larvae that did not inherit a protected *Lp* copy due to the toxicity of the construct's *Lp* gRNAs.

In spite of the efficiency of *zpg-Cas9* for initial integration, this first *Saglin* GD (termed *SagGD^{zpg}*) showed disappointingly modest homing levels at both the *Saglin* and *Lp* loci in the G2 generation

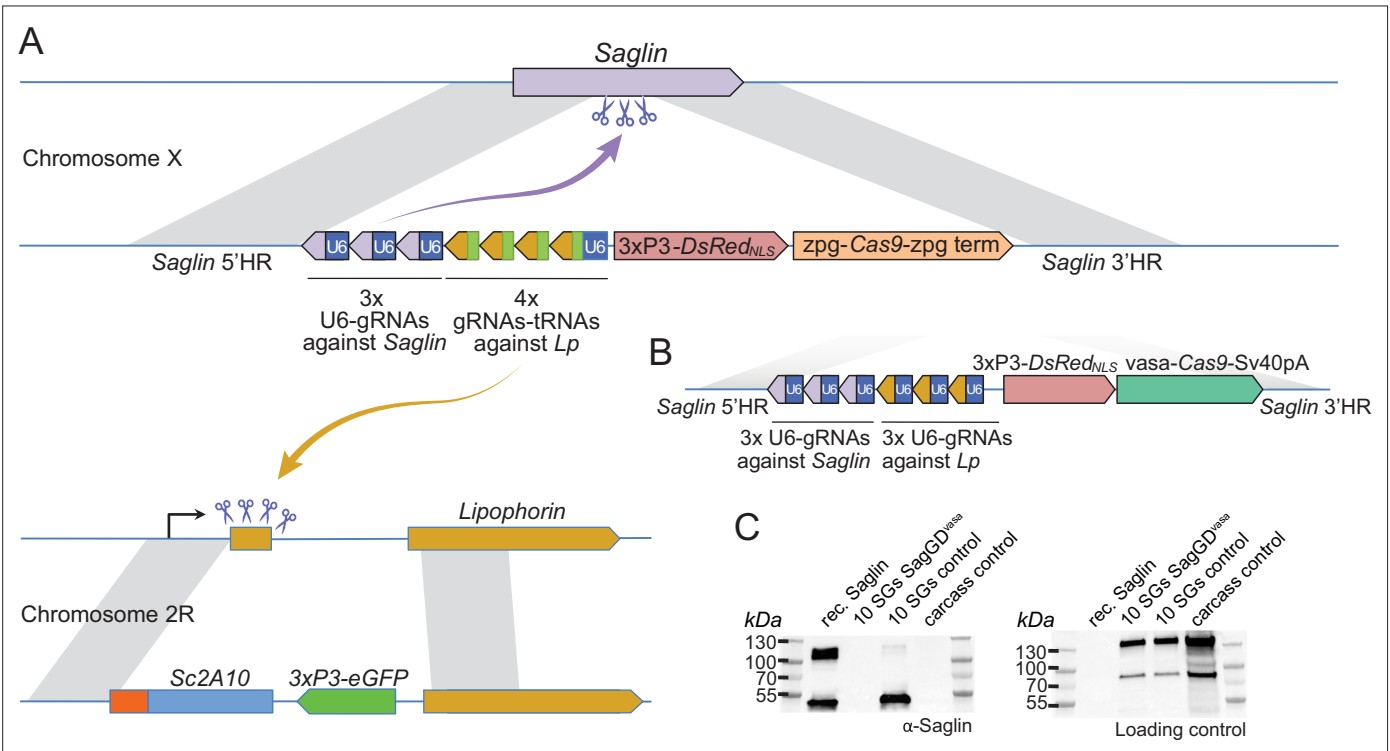

**Figure 5.** Designs of *Saglin*-based gene drives that also promote *Lp* locus modification and scheme of the chromosome conversion process. (**A**) Gene drive cassette comprising *Cas9* (under control of the *zpg* promoter) and an array of 7 gRNA-coding modules, inserted disruptively in the endogenous *Saglin* open reading frame on chromosome X along with a *3xP3-DsRed_{NLS}* fluorescence marker used to track the genetic modification. Three gRNAs, each expressed under control its own U6 promoter, target wild-type Saglin (purple arrow) and promote homing of the gene drive cassette. Four gRNAs separated by a repeated glycine tRNA are expressed under control of one U6 promoter, and target wild-type *Lipophorin* on chromosome 2 R (symbolized by yellow arrow) to promote *Lp::Sc2A10* homing. (**B**) Updated *SagGD^{vasa}* gene drive construct comprising six gRNAs, each under control of its own U6 promoter, and Cas9 under control of the *vasa* promoter and SV40 terminator sequences. (**C**) Western-blot using Saglin antibodies showing the absence of Saglin protein in the salivary glands of dissected *SagGD^{vasa}* homozygous females. The same membrane was re-probed with serum from a human volunteer regularly bitten by mosquitoes, providing a loading control with salivary and carcass protein signals.

The online version of this article includes the following source data for figure 5:

**Source data 1.** Western-blot using Saglin antibodies showing the absence of Saglin protein in the salivary glands of dissected *SagGD^{vasa}* homozygous females (left image).

**Source data 2.** Western-blot using Saglin antibodies showing the absence of Saglin protein in the salivary glands of dissected SagGDvasa homozygous females.

**Source data 3.** Western blot using serum from a human volunteer regularly bitten by mosquitoes, providing a loading control with salivary and carcass protein signals.

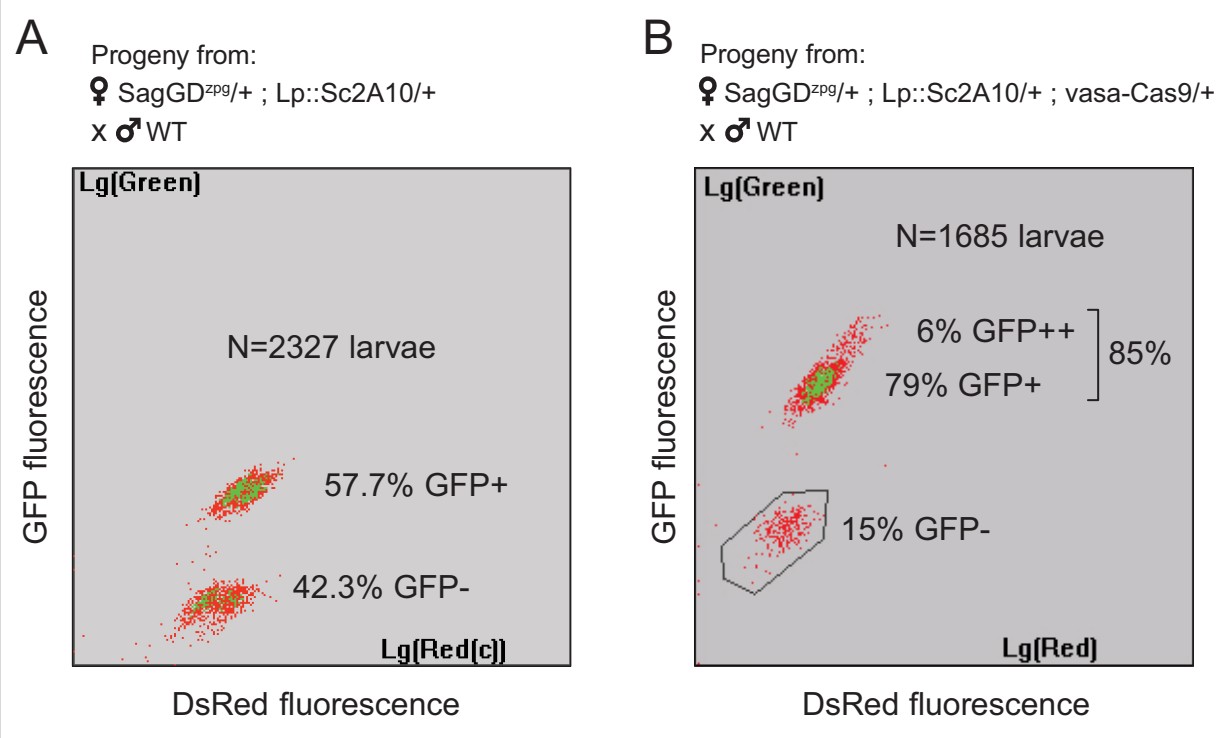

**Figure 6.** *Sag-GD^{zpg}* triggers modest homing at the *Lp* locus, which can be rescued by *vasa-Cas9*. (**A**) COPAS analysis of the progeny from [*SagGD^{zpg}*/+; *Lp::Sc2A*10/+] females crossed to WT. Weak DsRed fluorescence does not allow accurate separation of DsRed +and DsRed- larvae. Note that GFP inheritance is only slightly higher than the 50% expected in the absence of homing. (**B**) COPAS analysis of the progeny from triple transgenic [*SagGD^{zpg}*/+; *Lp::Sc2A*10/+; *vasa-Cas9*/+] females crossed to WT. Note the higher GFP homing rate. 6% of individuals appeared to be homozygous, revealing either unexpected homing in early embryos due to maternal Cas9 deposition, or accidental contamination of the cross with a few transgenic males. DsRed positive and negative larvae were indistinguishable with the COPAS settings used for this experiment.

(*Figure 6A*). Using COPAS flow cytometry, we examined GFP and DsRed inheritance in 2327 larvae that arose from the pooled G1 females crossed to WT males. It showed 57.7% GFP inheritance (*Lp::Sc2A10* locus) compared to 50% expected in the absence of homing. DsRed inheritance was more difficult to assess, as the low intensity of red fluorescence did not allow accurate gating to separate red fluorescent from negative larvae on COPAS diagrams. Approximate gating completed with visual examination of a sample of 200 larvae suggested 54.1% DsRed inheritance at the *Saglin* locus, an even more modest inheritance bias than for GFP at the *Lp* locus. Mild homing could be explained by position effects affecting the activity of the *zpg* promoter, by a lack of recombination at the target loci, or by inefficiency of gRNAs. To distinguish among these possibilities, we generated females carrying single heterozygous copies of the three unlinked transgenes *SagGD^{zpg}*, *Lp::Sc2A10* and a non-fluorescent (puromycin resistance-marked) *vasa-Cas9*. The progeny of these triple-transgenic females crossed to WT males showed markedly better homing rates (>79% GFP inheritance) (*Figure 6B*). Thus, *vasa-Cas9* was able to significantly rescue the modest homing rate of *zpg-Cas9* from *SagGD^{zpg}*, resulting in an efficient split GD. This suggests that the cloned *zpg* promoter sequence is sensitive to positional effects, being poorly active when inserted in *Saglin* or *Lp* in contrast to other previously reported host loci such as *DSX* or *Nudel* (*Kyrou et al., 2018*; *Hammond et al., 2021*).

We maintained one mosquito population of *Lp::Sc2A10* combined with *SagGD^{zpg}* (initial allele frequencies: 25% and 33%, respectively) and measured genotype frequencies after seven generations. This showed an increase in the frequency of both alleles (G7: *GFP* allelic frequency = 59.2%, phenotypic expression of *DsRed* in >90% of larvae, n=4282 larvae), indicating that the modest GD strength of *SagGD^{zpg}* was still able to increase the frequency of the *SagGD^{zpg}* allele over time and counteract the *Lp::Sc2A10* fitness cost.

**Table 2.** The *SagGD*^vasa^ GD shows high homing rates in the G2 generation at both the *Lp* and *Saglin* loci.

Six individual *SagGD*^vasa^ G1 females mated to WT males oviposited in individual tubes and their larval progeny was scored visually for GFP and DsRed fluorescence. Homing rates are calculated as the percentage of WT chromosomes converted to transgenic, i.e.: ((inheritance rate)–50%)x2.

| Female # | Total larvae | Negatives | GFP + only | DsRed + only | GFP + DsRed + | GFP inheritance and homing rate (*Lp* locus) | DsRed inheritance rate, homing rate (*Saglin* locus) |
|---|---|---|---|---|---|---|---|
| 1 | 89 | 0 | 0 | 0 | 89 | 100 % | 100 %, 100% |
| 2 | 52 | 0 | 4 | 0 | 48 | 100 % | 92.3%, 84.6 % |
| 3 | 59 | 0 | 0 | 0 | 59 | 100 % | 100%, 100 % |
| 4 | 59 | 0 | 6 | 0 | 53 | 100 % | 89.8%, 79.6 % |
| 5 | 50 | 0 | 5 | 0 | 45 | 100 % | 90%, 80 % |
| 6 | 57 | 0 | 3 | 0 | 54 | 100 % | 94,7%, 89.4 % |

## Driving Lp::Sc2A10 distantly from *Saglin* with a vasa-Cas9 gene drive

In order to achieve a stronger GD that may push *Lp::Sc2A10* to fixation, we re-built the *Saglin* GD construct, replacing *zpg* with the *vasa* promoter to control *Cas9* expression (*SagGD*^vasa^), in spite of this promoter being known for causing maternal deposition of Cas9 mRNA and/or protein, potentially resulting in undesired zygotic and somatic mutation at gRNA target loci.

We initially made two versions of *SagGD*^vasa^: one carrying the complex array of seven gRNAs described above for *SagGD*^zpg^ (as in **Figure 5A**); another carrying six tandem U6-gRNA units (**Figure 5B**; sequence provided in **Supplementary file 1E**). Both yielded similar results in early generations of the new mosquito lines. We focused all subsequent analyses on the latter version expressing three gRNAs against *Saglin* and three against *Lp*, because its fluorescent marker (*3xP3-DsRed*) proved easier to track in neonate larvae than *OpIE2-DsRed* marking the tRNA-containing version (the *OpIE2* promoter integrated in the *Saglin* locus only became active late in larval development).

About 35 surviving G0 injected male mosquitoes (which were homozygous for *Lp::Sc2A10* to prevent any premature damage on *Lp*), outcrossed to WT females, yielded about 20 DsRed positive G1 females and no positive male, again an indication of proper insertion of the GD cassette onto the X chromosome, where *Saglin* is located. These G1 females (heterozygous at both *SagGD*^vasa^ on the X chromosome and *Lp::Sc2A10* on the second chromosome) developed normally to adulthood and were fertile when crossed to wild-type males. High homing rates at both loci were recorded in G2 by examining the progeny of the individual G1 females backcrossed to WT males, a large majority of progeny inheriting both a *SagGD*^vasa^ copy (DsRed_{NLS}) and a *Lp::Sc2A10* copy (GFP) (**Table 2**).

We then sought to examine the homing rates at the two loci at a larger scale, using COPAS flow cytometry to analyse the progeny from [*SagGD*^vasa^/Y; *Lp::Sc2A10*/+] males backcrossed *en masse* to wild-type females, and from [*SagGD*^vasa^/+; *Lp::Sc2A10*/+] females backcrossed *en masse* to wild-type males. The male cross yielded only 405 larvae, of which 391 were GFP positive (GFP inheritance: 96.5%, corresponding to a homing rate of 93%). *DsRed* on the X chromosome was, as expected, passed on to 50% of the progeny (daughters). The female cross yielded 5197 larvae showing 95.1% *GFP* inheritance and 80.7% *DsRed* inheritance, corresponding to homing rates of 90.2% at the *Lp* locus and 61.4% at the *Saglin* locus.

Thus, in contrast to *zpg*, the *vasa* promoter was highly active in germ cells when inserted in the *Saglin* locus. The selected gRNAs, or homing at the *Lp* locus, appeared to be more effective than at the *Saglin* locus. Efficient homing lifted doubt on the activity of *Lp* and *Saglin* gRNAs in the GD construct, or on the propensity of the *Saglin* locus to undergo homologous recombination.

To verify that *SagGD*^vasa^ abolished *Saglin* expression, we dissected salivary glands from control and homozygous DsRed +mosquito females, and subjected them to western blotting using anti-Saglin antibodies. As expected, Saglin was undetectable in *SagGD*^vasa^ mosquitoes (**Figure 5C**).

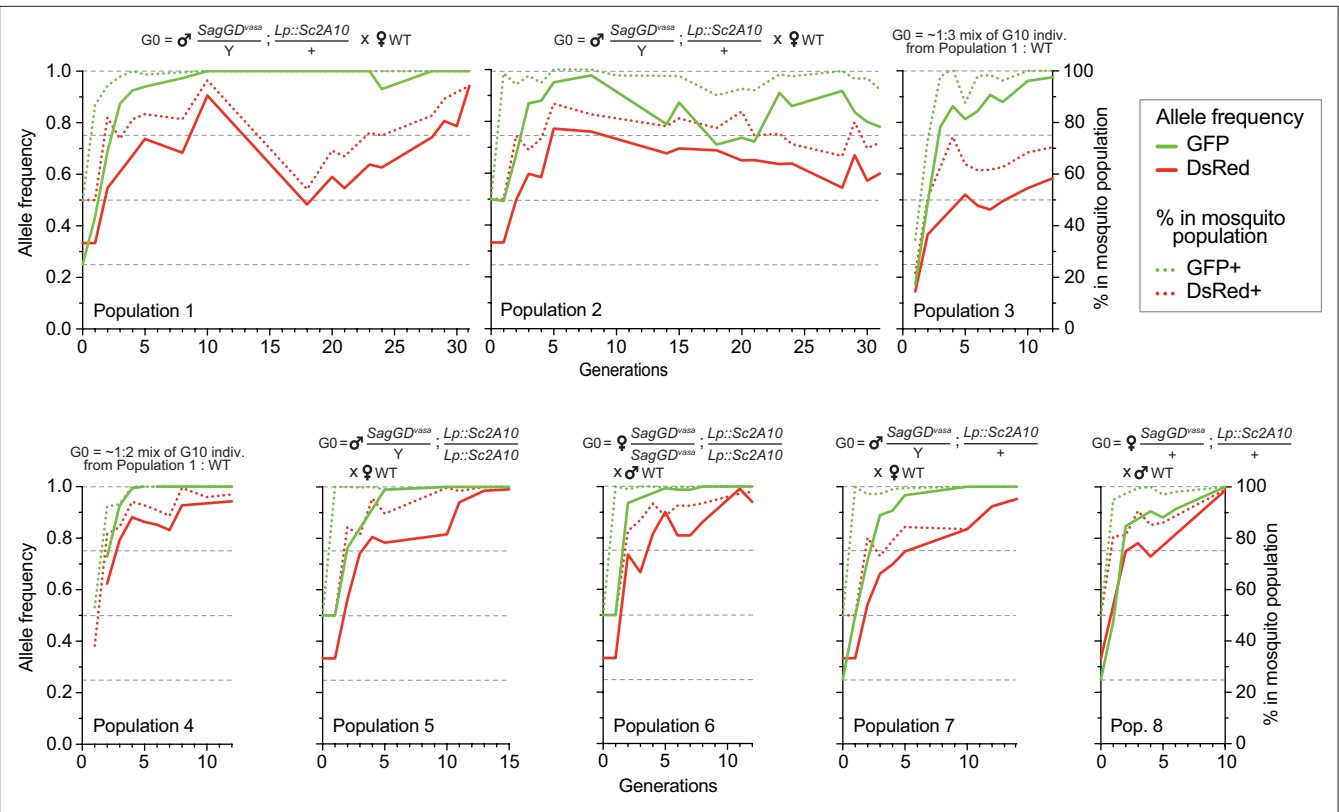

**Figure 7.** Temporal dynamics of the *Sag-GD^vasa* and *Lp::Sc2A10* transgenes in 8 mosquito populations. Transgenic mosquitoes carrying both transgenes were crossed to wild-type of the other sex (populations 1, 2, 5, 6, 7, 8) or mixed with wild-types (populations 3, 4) as indicated above the diagrams, and the frequency of each transgene in 1000–4000 neonate larvae of successive generations was tracked by flow cytometry (COPAS) using the GFP fluorescent marker of *Lp::Sc2A10* and the DsRed fluorescent marker of *Sag-GD^vasa*. Calculated transgene frequency, taking homozygous and heterozygous larvae into account (continous lines), and percentage of fluorescent mosquitoes (dotted lines) are shown on the graphs. For the autosomal transgene, frequency of each genotype in the population was calculated as([2 x(number of homozygotes)+number of heterozygotes]/ 2 x(total number of larvae)). For the X-linked transgene, frequency was ([2 x(number of homozygotes)+number of heterozygotes]/ 1.5 x(total number of larvae)). Numbers of larvae were obtained by gating corresponding clouds of larvae on COPAS diagrams (see *Supplementary file 5*) and recording the associated percentage measured by COPAS software, or by opening COPAS files and gating in WinMDI software. Gating was approximate due to partial overlap between different clouds of larvae. *Supplementary file 5* shows strips of all COPAS diagrams that served to extract this data, which provide a more accurate sense of transgene dynamics over generations.

## Temporal dynamics of the SagGD^vasa and Lp::Sc2A10 transgenes

We split the DsRed and GFP double-positive [*SagGD^vasa*/; *Lp::Sc2A10*/+] G2 males in two groups and crossed them to WT females, to establish two independent mosquito populations with a starting (G0) frequency of the autosomal *Lp::Sc2A10* transgene of 25%, whereas X-linked *SagGD^vasa* frequency was 33.3%. We maintained the two populations for >31 generations and, after 10 and 16 generations, derived six additional populations from the basal ones by outcrossing to WT. Populations 3 and 4 were established by mixing randomly selected transgenic mosquitoes (both males and females of generation 10) from populations 1 and 2, respectively, with wild-types, to mimic what may occur in a mixed-sex field release. Populations 5–8 were established by crossing single-sex transgenic mosquitoes to WT of the opposite sex, both to mimic a single-sex field release and to re-assess homing efficiency after 16 generations. To examine the long-term dynamics of both transgenes, we exploited GFP fluorescence from *Lp::Sc2A10* and DsRed fluorescence from *SagGD^vasa* to COPAS-analyze successive generations of all populations, with large sample sizes, over a long period of time (>2 years for populations 1 and 2). COPAS analyses allowed not only to distinguish negative from transgenic larvae but also, at least partially, heterozygotes from homozygotes for each transgene. Visual examination of the successive COPAS diagrams provides an accurate sense of the temporal dynamics of both transgenes generation after generation, and an appreciation of sample sizes (see snapshots of COPAS diagrams in *Supplementary file 5*). *Figure 7* offers a graphical representation of the same data, after extracting

approximate percentages of larvae of each genotype from the COPAS diagrams. In all populations, mosquitoes lacking *Lp::Sc2A10* disappeared rapidly, and all young populations rapidly consisted of a majority of homozygous *Lp::Sc2A10* mosquitoes (*Figure 7* and *Supplementary file 5*). Despite the absence of physical linkage between the two transgenes, DsRed-only individuals were strikingly absent from all early populations, indicative of the lethality of the *SagGD^vasa* transgene in the absence of a protected *Lp* allele. Importantly, toxicity of *SagGD^vasa* was due to its destructive action on the *Lp* locus rather than to a fitness cost of its disrupted *Saglin* host locus, since individuals carrying both DsRed and GFP thrived in all populations, and consistent with the reported absence of obvious fitness cost of *Saglin* mutants (*Klug et al., 2023*). Several populations (notably populations 1, 3, 4, 5, 6, 7) rapidly seemed fixed for *Lp::Sc2A10*. A trend for reversal, with *Lp::Sc2A10* progressively giving way to GFP negative individuals (presumably functional *Lp* mutants refractory to homing), was observed from G14 on for population 2, although GFP positive individuals still represented a large majority of the population (>90% from G1 on), while GFP allele frequency reached 0.98 at G8 before decreasing and oscillating between 0.71 and 0.92 after G14. For the *Saglin* locus, COPAS diagram interpretation must take into account the position of this gene on the X chromosome, with male mosquitoes bearing a single copy and appearing in the same larval cloud as heterozygous females (dosage compensation of the transgene not being apparent at this locus). Drive will be slower as it can occur only in females. While the majority of mosquitoes of all populations rapidly carried at least one copy of *SagGD^vasa*, we observed that a substantial fraction of DsRed negative individuals (up to 30%), likely carrying GD refractory *Saglin* mutants, persisted in some populations (*Figure 7* and *Supplementary file 5*).

## Detection of homing-refractory *Saglin* mutants

As a non-essential gene, *Saglin* is likely to accumulate mutations in evolving mosquito populations due to failed homing events of the GD construct associated with NHEJ repair at the three gRNA target sites. Indeed, DsRed negative mosquitoes (lacking the GD construct in *Saglin*) persisted at a relatively stable frequency of approximately 30% in populations 2 and 3 (*Figure 7*). To characterize *Saglin* mutants, we COPAS-extracted 750 and 150 DsRed-negative larvae from a surplus of larvae from generation 4 of population 1 and generation 3 of population 2. DsRed-negative individuals represented 18 and 31% of these populations, respectively, with the *Saglin* gene on their X chromosome having been exposed to between one and four rounds of Cas9 activity. We PCR amplified a *Saglin* fragment spanning the 3 gRNA target sites, and subjected the amplicon to high-throughput sequencing. While the majority of sequenced DsRed negative alleles still corresponded to wild-type *Saglin* having thus far escaped GD activity, a wide variety of *Saglin* mutations, different between the two sampled populations, were readily identified (*Figure 8a and b*). Interestingly, a majority of these *Saglin* haplotypes were mutated only at the target site of gRNA2, suggesting that this gRNA was the most active of the three and that Cas9, confronted to a pool of three *Saglin* and three *Lp* gRNAs, will not necessarily cleave all target sites in a given germ cell. However, combinations of mutations at gRNA1, gRNA2 and/or gRNA3 target sites were also identified. Several haplotypes were consistent with iterative action of Cas9 in lineages of haplotypes, a specific mutation occurring either singly, or associated to one or two additional mutations at the other gRNA target sites. Deletions spanning two gRNA target sites were rare in our samples (one instance of such a haplotype was detected, carrying a deletion between gRNA1 and gRNA2). Overall, Cas9 was not highly active at the *Saglin* locus, some target sites being missed at each generation and combinations of mutations accumulating iteratively as germ cells were exposed to Cas9 in successive generations. Therefore, the target locus of a multiplex gRNA GD experiencing failed homing at a given generation resulting in mutation of one target site may still undergo successful homing in later generations, as long as all gRNA targets are not yet mutated. Inexorably however, *Saglin* mutants fully refractory to the GD will form in mosquito populations; we recovered at least one haplotype with mutations in all three target sites after four generations (*Figure 8a*).

## Appearance of a GD-refractory Lp mutant

NHEJ mutants in *Lp* were expected to emerge at much lower frequency than *Saglin* mutants in evolving populations, given that *Lp* is an essential gene under strong selective pressure. While tracking the frequency of transgenes in [*SagGD^vasa*; *Lp::Sc2A10*] mosquitoes, we noted the appearance of a few GFP negative larvae in populations 2 and 3. These amounted to 2.4% of total larvae in generation 14

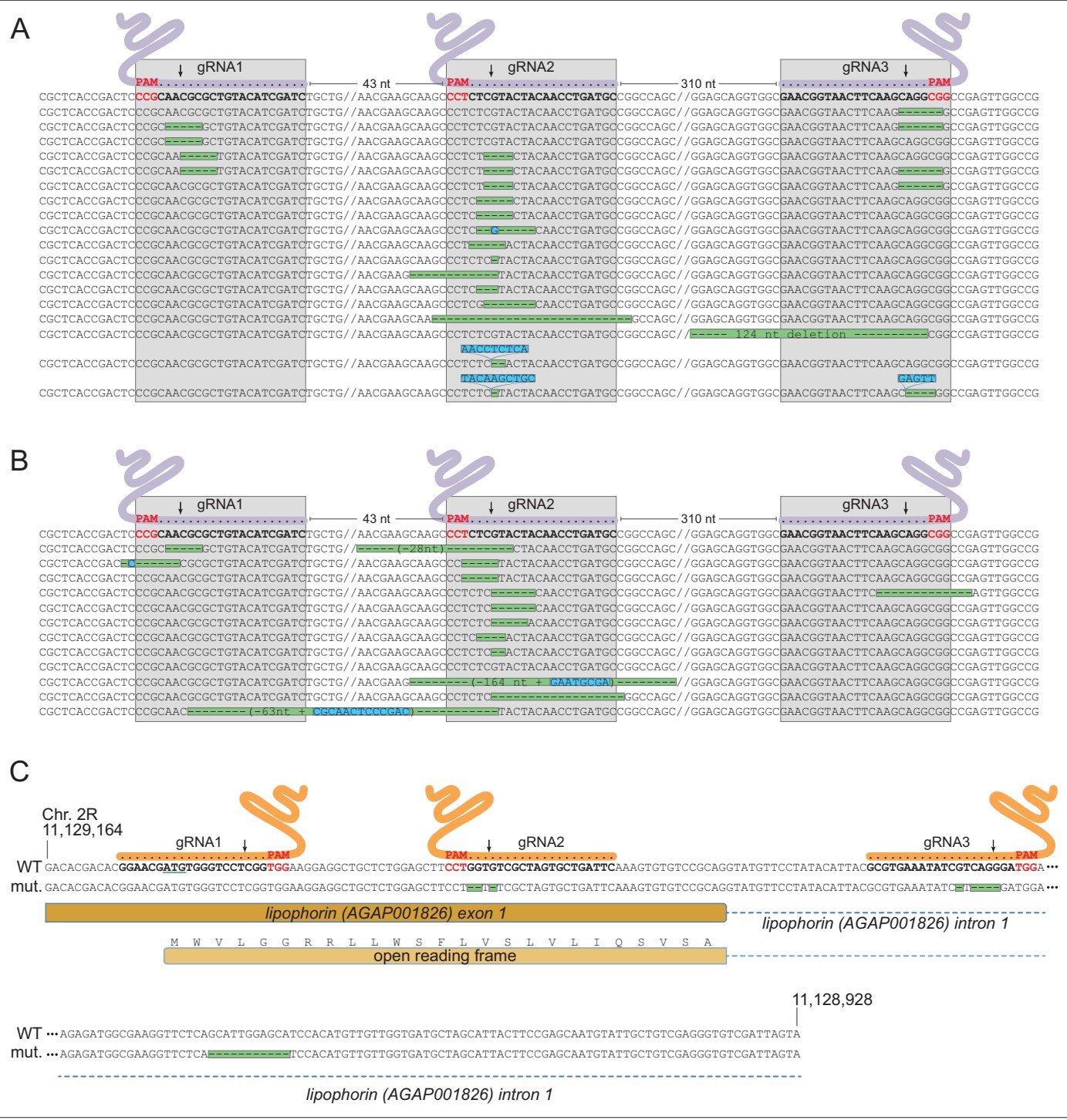

**Figure 8.** Characterization of mutations in target genes. (**A, B**) Characterization of *Saglin* failed homing mutations. A PCR product spanning the three *Saglin* gRNA target sites was amplified from 150 DsRed-negative larvae from generation 4 of population 1 (**A**) or from 750 DsRed-negative larvae from generation 3 of population 2 (**B**) and subjected to high-throughput amplicon sequencing. The different *Saglin* mutant haplotypes discovered in each sample are aligned to the WT sequence (top) with target site and protospacer-adjacent motifs (PAMs) indicated. Arrows point to Cas9 cleavage sites. Deleted nucleotides are highlighted in green, inserted nucleotides in blue. // indicates portions of WT sequence not represented on the figure. The distance between two gRNA target sites (number of nucleotides) is indicated above the WT sequence. (**C**) gRNA target sites and failed homing-induced mutations in the *Lp* gene. The WT sequence of the 5' region of the *Lp* gene shows exon and intron sequences, the position of gRNA target sites with PAMs (red), and the ATG initiator codon (underlined). Nucleotides shown in orange were deleted in twelve sequenced homozygous mutant mosquitoes. An additional, 11 bp deletion in the intron, distant from gRNA target sites (red) was unexpected.

of population 2. We COPAS-isolated a sample of these larvae and grew them to adulthood, expecting that they might die during development due to unrescued deleterious mutations in *Lp*. However, these GFP negative mosquitoes developed to healthy and fertile adults. Sequencing the *Lp* gene in 12 individuals revealed that all were homozygous for the same array of mutations within two of the three Lp gRNA target sequences (*Figure 8c*). One mutation deleted three nucleotides before the PAM of gRNA2, removing a single valine from the Lp signal peptide (MWVLGGRRLLWSFL<u>V</u>SLVLIQSVSA, missing valine underlined). The other is a 5-base deletion within the target of gRNA3 in the first *Lp* intron. Viability of the homozygous mutant indicates that this combination of mutations, while rendering *Lp* refractory to Cas9-mediated gene conversion, preserved *Lp* function. This suggested that (i) the first of the three gRNA expressed from the gRNA array did not act; and (ii) combinations of mutations preserving the function of target essential genes can emerge despite the cumulative activity of two gRNAs, which in this case was certainly facilitated by the fact that one of them targeted an intron, more permissive to change.

## Genetic instability of the gRNA array

The gRNA array in the *SagGD^vasa* construct being a repetitive tandem of six U6-gRNA units, genetic instability may be expected and observable after several generations of mosquito breeding. We examined transgene integrity in 48 individual mosquitoes of the 32nd generation after transgenesis, by genotyping DsRed-positive males, which possess a single gene drive copy on their X chromosome. PCR primers were chosen to span the gRNA array. Resulting amplicons were examined on agarose gels (*Figure 9*) and sequenced. Interestingly, two derivative alleles of the gene drive construct dominated the four sampled mosquito populations: a longer version that still retained five of the original six gRNAs, and a shorter version that retained only two gRNAs. The original transgenesis vector used for microinjection was resequenced and verified to contain all six gRNA expression units; therefore, deletions occurred post integration in the mosquito genome. gRNA deletions probably began to occur early in the evolution of the transgenic populations, since the same deletion of the first *Lp* gRNA was detected in both populations 1 and 2, separated at the third generation following transgenesis and subsequently bred independently for 29 generations before sampling. The larger deletion of 4 gRNAs was not observed among the 12 sampled males from population 2, whereas it was the most frequent allele in population 1 and its two daughter populations 5 and 7. This larger deletion may have arisen from a further loss of gRNA units in the allele carrying 5 gRNAs, since *Lp* gRNA1 is missing in both alleles, its position occupied by *Lp* gRNA3 (*Figure 9*). The early loss of *Lp* gRNA1 likely explains the lack of any mutation at its target site in the homing-refractory *Lp* mutant identified above. Regardless of the number of missing gRNA units, the *Cas9* part of the transgene appeared to be intact in these mosquitoes (*Figure 9C*). Interestingly, both deleted versions in principle retain the capability to promote homing at both *Saglin* and *Lp* target loci, as even the short form carries one active gRNA against each gene. However, since the single remaining *Lp* gRNA is targeting the *Lp* intron, this shorter GD derivative would easily select for novel homing-refractory *Lp* mutants and this derivative gene drive is likely to be much less durable.

## *Vector competence of [*SagGD^vasa*; Lp::Sc2A10] mosquitoes*

We expected that the combination of Sc2A10 expression with *Saglin* loss-of-function would enhance the *Plasmodium* transmission-blocking phenotype. To assess this, we mixed *[SagGD^vasa*; Lp::Sc2A10]* mosquito larvae from the 20th generation of population 2 with an equal number of non-fluorescent wild-types. The resulting cage of adults was blood-fed on a mouse infected with *Pb-PfCSP*^hsp70-GFP*. Seventeen days later, females carrying visible GFP parasites were selected and split into six cups of 10 transgenic, and four cups of 10 WT. All four mice exposed on day 18 to the infected control females developed parasitemia, whereas all six mice exposed to infected transgenics remained negative (*Supplementary file 4D*). Six additional mice were exposed to 8–12 bites from *[*SagGD^vasa*; Lp::Sc2A10]* oocyst-carrying females of the 4th generation of population 6, corresponding to the 23rd generation since transgenesis. Only two of these six mice developed parasitemia 6 days after infection (*Supplementary file 4E*). While the small scale of these experiments cannot strictly confirm whether the *Saglin* knockout enhanced protection against transmission (especially as in these experiments we discarded less-infected mosquitoes, which are expected to result from *Saglin* loss-of-function), the absence of infection in a total of 10 out of 12 mice, in contrast to 4 mice out of 4 infected when

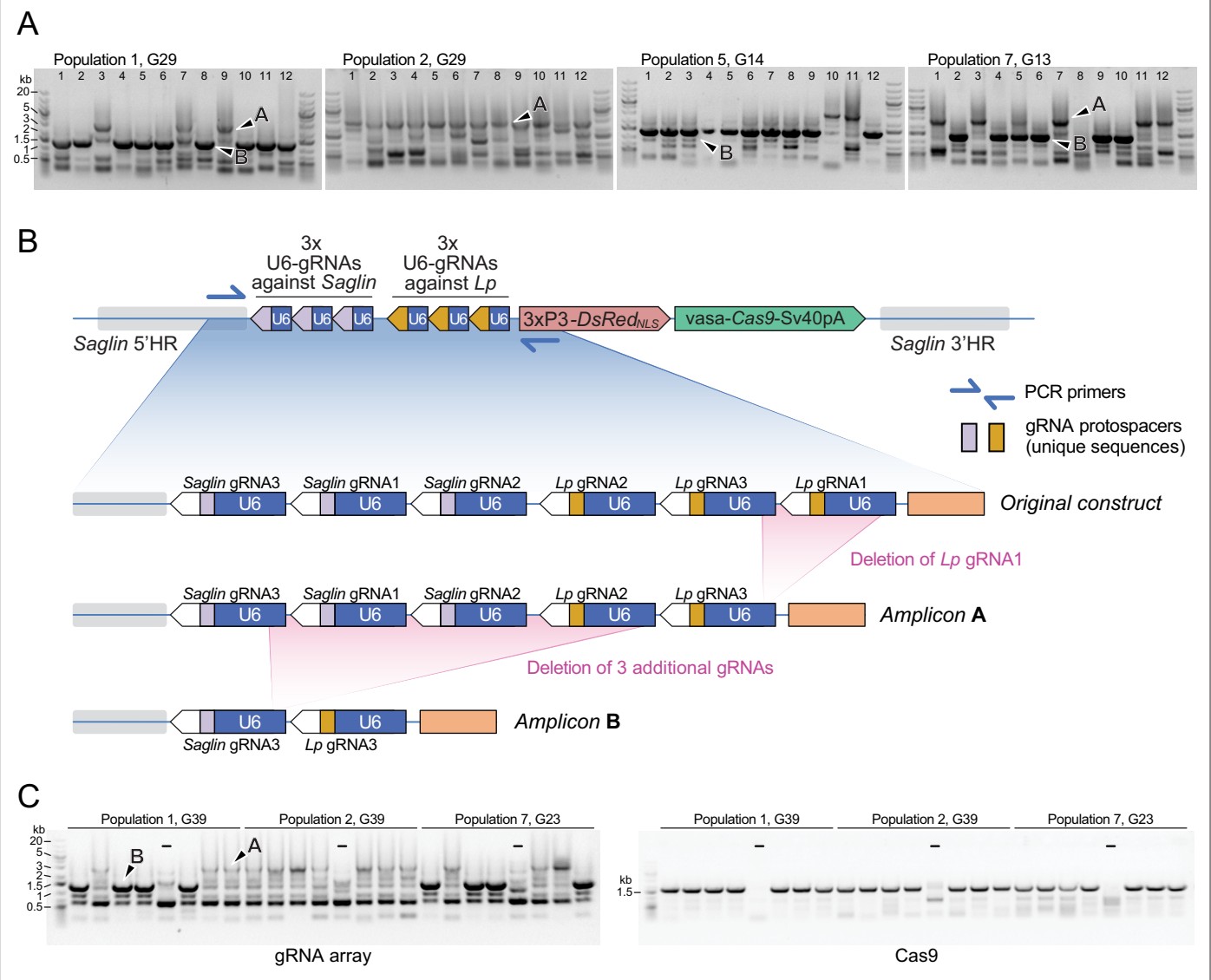

**Figure 9.** PCR genotyping of 31st generation individual *Sag-GD^vasa* mosquitoes reveals deletions in the gRNA array. (**A**) PCR amplicons spanning the six initial gRNAs in *Sag-GD^vasa* were generated from 12 individual male mosquitoes (carrying a single transgene copy) from each of populations 1, 2, 5, and 7, 31 generations after initial transgene integration (corresponding to the indicated generation number for each tracked population) and resolved on agarose gels with 1 kb +DNA ladder (Thermofisher) as a size marker (upper panels). Arrows point to examples of PCR products specific of the transgene as confirmed by Sanger sequencing; other bands are PCR artefacts. Sanger sequencing of amplicons of types A and B showed that they contain only five and two residual gRNA coding units, respectively, as schematized in panel (**B**). (**C**) The same deletions were detected again 10 generations later (left panel). PCR primers amplifying *Cas9* were used in parallel (right panel), to verify the presence of the *Cas9* part of the transgene. '–' signs indicate control PCRs performed on wild-type genomic DNA. Sample loading order is identical in the two panels.

The online version of this article includes the following source data for figure 9:

**Source data 1.** The top part and the bottom part of each gel image was cropped to exclude empty space and re-arranged horizontally to generate *Figure 9A*.

**Source data 2.** The top part and the bottom part of each gel image was cropped to exclude empty space and re-arranged horizontally to generate *Figure 9A*.

**Source data 3.** The top part and the bottom part of each gel image was cropped to exclude empty space and re-arranged horizontally to generate *Figure 9A*.

**Source data 4.** The bottom parts of these gel images was cropped to constitute the left and right panels of *Figure 9C*.

**Source data 5.** The bottom part this gel image was cropped to constitute the left panel of *Figure 9C*.

**Source data 6.** The bottom part this gel image was cropped to constitute the right panel of *Figure 9C*.

exposed to infected WT mosquitoes (**Figure 2D**), showed that *[SagGD^vasa; Lp::Sc2A10]* mosquitoes maintained a markedly decreased ability to transmit *Plasmodium* after >20 generations of existence.

## Discussion

We hijacked an essential gene, *Lipophorin* (*Lp*), to force *Anopheles coluzzii* mosquitoes to co-express an anti-malarial factor. The choice of the *Lp* gene was motivated by its exceptionally strong expression level (Lp being the most abundant constitutively expressed protein in mosquito hemolymph) and by its endogenous signal peptide, which we could exploit to obtain concomitant secretion of Sc2A10 with Lp protein. The existence of an endogenous proteolytic cleavage site within the Lp protein, that we duplicated to detach Sc2A10 from Lp, was also advantageous, as well as the natural intron immediately following the signal peptide-encoding exon 1 that could accept a visual selection marker for easy tracking of the genetic modification. This approach is related to the integral gene drive strategy (*Hoermann et al., 2021*) in that anti-malarial factors are expressed together with an endogenous gene, but here the modified endogenous gene does not comprise any component of a gene drive —in the case of integral gene drives, gRNAs are inserted in a synthetic intron. The genetic modification does not disrupt expression or function of the *Lp* essential gene. On the contrary, Lp protein expression is intended to remain as unaffected as possible, but obligatorily proceeds through translation of the Sc2A10 antimalarial factor inserted between the Lp signal peptide and the rest of the Lp protein. This design minimizes the risk of spontaneous transgene loss by mutation, since frameshift or nonsense mutations in the *Sc2A10* coding sequence would prevent *Lp* expression, resulting in their elimination by natural selection. Furthermore, the absence of repeated motifs in the synthetic sequence decreases the risk of in-frame deletions. The possibility of a loss of the transgene's anti-*Plasmodium* activity should therefore be limited to missense mutations or in-frame deletions arising in the *Sc2A10* coding sequence, or to *Plasmodium* evolution, notably under selection pressure imposed by the transgene itself (*Marshall et al., 2019*). However, the 2A10 antibody targets a highly repeated epitope on the CSP protein (the NANP motif), that, in addition to being repetitive, is also conserved across *P. falciparum* strains. A combination of mutations leading to the complete loss of this epitope in *Plasmodium* therefore seems unlikely, and would probably cause a dramatic loss of parasite fitness as CSP is required in several processes in the sporozoite's journey between two hosts (*Coppi et al., 2011*; *Balaban et al., 2021*).

Hijacking Lp and its signal peptide is an approach that could be widely applicable to other possible factors acting in the hemolymph to attack various parasite stages, such as ookinetes, which represent a particularly vulnerable stage of parasite development in the mosquito. To make mosquitoes that express and combine additional secreted factors against parasites, *Vitellogenin* (AGAP004203) could also be engineered in the same manner, taking advantage of its own signal peptide, extremely high expression level but also transient induction following a blood meal. Blood meal inducibility would be particularly suitable for antimalarial factors specifically targeting ookinetes, as they emerge on the hemolymph-bathed side of the midgut exactly within the short time window when *Vitellogenin* is expressed. Mosquitoes would not be exposed during their development to a factor expressed from *Vg*, limiting potential fitness costs. As proposed by *Hoermann et al., 2021 Carboxipeptidase* (*Cp*) represents an additional attractive host locus for antimalarial factors, sharing many of the advantages of *Vg*. A single GD expressing multiplexed gRNAs could stimulate the simultaneous spread of *Lp, Vg* and *Cp* modified, anti-parasitic alleles.

In spite of the high Lp protein secretion in mosquito hemolymph, we only detected less than 1/10 of this molecular amount for Sc2A10 in hemolymph by mass spectrometry. Apart from technical limitations in detection sensitivity (e.g. biased trypsin digestion of some peptides) this could be explained by faster removal from the hemolymph and /or faster degradation of Sc2A10 compared to Lp. Different, more stable anti-*Plasmodium* peptides that could accumulate to higher levels may show even better efficacy as transmission-blocking agents compared to Sc2A10. This also highlights the benefit of choosing a strong promoter such as that of *Lp* to ensure maximal production of antimalarial effectors in the hemolymph. A single copy of the *Lp::Sc2A10* transgene still showed a reduction in transmission. Of note, the rodent model of malaria used here (*P. berghei* expressing CSP from *P. falciparum*) is characterized by very high infection rates, typically producing dozens of oocysts per midgut, each of which can release hundreds of sporozoites in mosquito hemolymph. The ability of secreted Sc2A10 to decrease sporozoite transmission even in this high infection context is highly encouraging,

given the much lower levels of *P. falciparum* (typically 1 to a few oocysts) in field infections. An even greater anti-*Plasmodium* activity of Sc2A10 may be reached by fusing it to antimicrobial peptides such as CecropinA, as achieved by Isaacs et al. in conventional transgenic *An. stephensi* mosquitoes (*Isaacs et al., 2011*; *Isaacs et al., 2012*) and subsequently in the first published GD for *An. stephensi* (*Gantz et al., 2015*). In these studies, the Sc2A10::CecropinA fusion was expressed intermittently under the control of the blood-meal inducible *Vitellogenin* promoter, and combined with a second scFv targeting *Plasmodium falciparum* ookinete Chitinase 1 also expressed after blood meals under the control of the *Cp* promoter. While the insertion of *Sc2A10::CecropinA* within *Lp* coding sequences may further improve the efficiency of our approach, there is a risk that CecropinA expressed at all developmental stages, and an increased complexity of the Lp fusion protein, worsen the fitness cost already observed in *Lp::Sc2A10* mosquitoes.

To force the spread of the *Lp::Sc2A10* transgene in mosquito populations, we tested several gene drive (GD) designs. A same-locus suppression drive was also installed in the *Lp* gene, creating a loss-of-function allele. The *Lp::Sc2A10* allele was intended to act as an R1-type, gene drive refractory functional allele (*Champer et al., 2018*) that would supplant the suppression gene drive and invade the population. This was unsuccessful, as *Lp* proved largely haplo-insufficient. In contrast, distant-locus GDs disrupting *Saglin*, a non-essential but pro-parasitic gene, were efficient at promoting invasion of both the *Saglin*, and particularly the *Lipophorin*, modified alleles. Unlike the first approach, this design may allow Cas9 and gRNA-coding genes to persist indefinitely within the invaded mosquito population (unless nonfunctional resistance alleles outcompete the drive allele in the long run). Although for this reason we initially preferred the same-locus approach in which *Lp* function disruption by the insertion of CRISPR/Cas9 components caused their gradual disappearance by natural selection, the added benefit of a GD installed in *Saglin* is that it should further reduce the transgenic mosquitoes' vectorial capacity, a combination of two distinct mechanisms to decrease sporozoite transmission being more robust to combat *Plasmodium* durably.

Importantly, we observed that same-locus or distant-locus constructs carrying guide RNAs that target wild-type *Lp* are inviable, and rapidly go extinct in the absence of a recoded *Lp* allele such as *Lp::Sc2A10* (or spontaneous *Lp* mutants immune to GD that may arise in GD populations). The genetic design established here can be viewed as a two-component GD in which one component (*Lp::Sc2A10*) has no drive capacity by itself but (i) rescues the lethality, and (ii) hitchhikes on the drive capacity of the other (*SagGD$^{vasa}$*) ensuring its own hyper-Mendelian spread. The two transgenic loci are, therefore, reciprocally dependent on each other: *Lp::Sc2A10* depends on *SagGD* for its long-term persistence and spread in a population, and *SagGD* depends on *Lp::Sc2A10* as a rescue allele of the essential *Lp* target for its survival. This design can be seen as a two-locus variation of rescue-type GDs (*Adolfi et al., 2020*; *Champer et al., 2020a*).

Because *Saglin* knockout mutants bear no fitness cost at least in laboratory conditions, GD-refractory *Saglin* alleles generated by failed homing events are certain to accumulate over time (all the more so as their lack of toxicity toward the WT *Lp* allele confers them a high selective advantage compared to *Sag*GD). If the demise of the *Saglin* GD occurs sufficiently late, *Lp-Sc2A10* could still reach high levels or even fixation in a mosquito population. If, however, GD refractory mutations in *Saglin* accumulate rapidly and no longer support *Lp::Sc2A10* maintenance, the elimination of both *SagGD$^{vasa}$* and *Lp::Sc2A10* will be inexorable. Thus wild-types alleles are likely to re-establish at both loci once the frequency of active *Saglin* GDs drops, especially under conditions of strong mosquito immigration from regions adjacent to the initial GD spread range. Still, even as genetic resistance to the GD develops, the resulting high frequency of *Saglin* loss-of-function mutant alleles arising from failed homing events, with their transient selective advantage while *SagGD$^{vasa}$* and wild *Lp* alleles co-exist, should decrease global vector competence.

Once a GD installed in the *Saglin* locus is defeated by the accumulation of GD-refractory mutations, new generations of GD installed at different loci could be launched to re-ignite the spread of the *Lp::Sc2A10* allele. For instance, GDs built for the same mosquito species in the frame of other projects could also incorporate gRNAs re-igniting the spread of *Lp::Sc2A10*, promoting antimalarial approaches similar to combination therapies. In this respect, transgenic systems permitting the expression of multiplexes of many gRNAs would be instrumental. Some of the constructs presented here incorporate multiplex gRNAs combining individual U6-gRNA cassettes with a U6-gRNA::tRNA quadruple gRNA expression cassette, with the drawback of displaying many repeated motifs (U6

promoters, multiple units of tRNAglycine, gRNA invariable region). Genetic instability of these constructs, with loss of gRNA expression segments, was prominent in our experiments. This is illustrated by the high prevalence, in our tracked mosquito populations, of a GD derivative allele carrying only two residual U6-gRNA expression units, from the original tandem of six. Luckily, this deletion allele still retains driving potential for both the *Saglin* and *Lp* loci, but it would take only one additional recombination event to reduce the array to a single gRNA. This would result either in a *Saglin*-only, non-lethal, GD allele, which may continue to reduce vectorial capacity of the mosquito population by mutating *Saglin*, or in a non-driving *Saglin* mutant still able to promote some *Lp::Sc2A10* spread by a split GD effect (but particularly able to generate *Lp* R1-type mutants given the intronic target of the single residual gRNA). We are currently working to improve gRNA multiplex design by increasing the number and variety of tRNA spacers, so that the 76 bp constant region of each gRNA would be the only remaining repeated element. To further optimize future GD design, modeling studies can now aid in determining the optimal number of gRNAs in a multiplex, depending on the specific GD design and purpose (*Champer et al., 2020b*).

In addition to this and to the stabilization of multiplex gRNA arrays, other paths to improvement of the system presented here include avoiding gRNAs targeting intronic sequences, and the use of promoters better restricted to the germline to control Cas9 expression and limit GD refractory mutation emergence. Interestingly, the *zpg* promoter that in theory possesses this advantage showed low activity once inserted in the *Lp* and *Saglin* loci, in contrast to the *DSX* or *Nudel* loci (*Hammond et al., 2016*; *Kyrou et al., 2018*). This suggests that the *zpg* promoter is more sensitive to the local chromatin context than the *vasa2* promoter, similarly to the *sds3* promoter in *Aedes aegypti* (*Anderson et al., 2023*), and that its usability in GD constructs may be restricted to a subset of genomic loci. The *nanos* promoter (*Gantz et al., 2015*; *Terradas et al., 2022*; but see *Hammond et al., 2021*) may represent an interesting alternative. Finally, tighter clustering of gRNA target sites at target homing loci, especially *Saglin*, should improve gene drive performance by reducing the length of DNA sequences flanking the cut site that bear no homology to the repair template on the sister chromosome and need to be resected by the repair machinery to allow homing (*López Del Amo et al., 2020*).

A simpler design to promote the spread of antimalarial factors such as Sc2A10 in mosquito populations would be a single-locus GD (for example installed in *Saglin* or other pro-parasitic genes if the idea of consolidating the anti-*Plasmodium* activity via two mechanisms is retained). Few loci, however, can provide the benefits offered here by *Lp* that include a suitable expression pattern, high expression, an endogenous secretion signal, endogenous proteolytic cleavage, a natural intron ideally positioned to host a fluorescent marker, and the status of essential gene that a translational fusion exploits to limit spontaneous transgene loss by mutation. Besides, adding the *Cas9* and gRNA-coding cassettes to an intron of an essential gene chosen as GD host would add >5 kb of foreign DNA sequence, thereby significantly increasing the complexity of the modification and the risk of affecting functionality of the essential gene. For example, inadvertent intron splice sites in synthetic sequences can preclude proper mRNA splicing (*Hoermann et al., 2022*); we also encountered this problem when inserting a different anti-*Plasmodium* ScFv in *Lp* (*Green, 2019*).

In conclusion, the *Sag-GD^vasa^* transgene presented here constitutes a viable GD only in the presence of an engineered *Lp* allele, which by itself has no drive capacity but will hitchhike with *Sag-GD^vasa^* to spread at hyper-Mendelian rates. In caged mosquito populations, both genetic loci initially spread together and decrease mosquito capacity to transmit *Plasmodium* parasites expressing CSP from *P. falciparum*. Mutations at both loci, especially in *Saglin* that is under low selection pressure, arise due to failed homing events, have a selective advantage over the toxic *SagGD^vasa^* allele, and over time should replace the GD, so that the system carries the seeds of its own removal. The anti-*Plasmodium Lp::Sc2A10* allele will likely slowly disappear due to its fitness cost, unless its spread is re-ignited by updated distant-locus GDs. Although such a reversibility could be seen as a safety advantage for modification GDs (and may be enhanced by intentionally releasing laboratory-selected resistance alleles), future work should focus on establishing GD constructs in *Saglin* that limit the emergence of GD refractory alleles to prolong their potential field lives.

# Materials and methods

## Plasmid construction

We used Golden Gate Cloning (*Engler and Marillonnet, 2014*) to assemble all parts of the *Lp* (AGAP001826) knock-in plasmid shown in *Figure 1B* in destination vector pENTRR4-ATCC-LacZ-GCTT (Addgene# 173668). The full annotated plasmid sequence is provided in *Supplementary file 1B*. The recoded beginning of *Lp*, the Sc2A10 coding sequence and intronic 3xP3-GFP-Tub56D marker cassette were ordered as synthetic DNA gBlocks (IDT DNA, Belgium) flanked by appropriate *Bsa*I restriction sites. The Lp endogenous Furin cleavage site (RFRR) was replicated along with 3 N-terminal and 4 C-terminal flanking amino acids between the Sc2A10 and ApoLpII amino acid sequences. This, and retaining 7 endogenous nucleotides upstream of the intron 5′ splice junction after the *Sc2A10* coding sequence, resulted in the addition of 7 non-natural amino acids (GIRESAA) to the N-terminus of the ApoLpII moiety. 1520 bp of 5′ and 1027 bp of 3′ flanking sequence were cloned on either side of the modified *Lp* sequence to be knocked-in. Adjacent to the 3′ *Lp* homology arm, we cloned three guide RNA (gRNA) expression modules (U6 promoter — gRNA coding sequence — terminator) that specify Cas9 cleavage within the first exon and intron of the genomic (but not recoded) *Lp* sequence. The three gRNA-expressing modules were prepared in pKSB-sgRNA1—3 (Addgene #173671—173673) as described (*Dong et al., 2018*) with gRNAs targeting the motifs: GAATCAGC ACTAGCGACACCAGG, GCGTGAAATATCGTCAGGGATGG and GGAACGATGTGGGTCCTCGG TGG (PAMs underlined) in *Lp*. One gRNA target site (with PAM) was included on the 3′ flanking homology arm's distal extremity to favor linearization of the donor plasmid after egg microinjection. This donor plasmid was called *pENTR R4 3xgRNA Lp::Sc2A10*. Plasmids encoding gene drive components for insertion into the *Lp* and *Saglin* sequences were assembled by Golden Gate Cloning in the same manner, including *promoter-Cas9-terminator* and *3xP3* or *OpIE2-DsRed$_{NLS}$*-SV40 modules, the sequences of which are provided in the complete plasmid sequence (*Supplementary file 1A-D*). The first version of the *zpg* promoter was PCR amplified from *A. coluzzii* genomic DNA with primers GGTC TCtcagcgctggcggtggggac and GGTCTCccattctcgatgctgtatttgttgttgggctgTttgtta, *zpg* terminator with GGTCTCCaattGaggacggcgagaagtaatcata and GGTCTCggatatcgcataatgaacgaaccaaagg, and cloned to make *Bsa*I Golden Gate cloning modules. The *zpg-Cas9* cassette in *SagGD$^{zpg}$* was subcloned from p17410 (*Kyrou et al., 2018*). The *vas2* version of the *vasa* promoter (*Papathanos et al., 2009*) was re-amplified from *A. coluzzii* genomic DNA with primers CggtctcaATCCcgatgtagaacgcgagcaaa and CggtctcaCATAttgtttcctttctttattcaccgg to make a *Bsa*I Golden Gate cloning module.

## *Cas9*-expressing mosquito strains

Three transgenic mosquito strains with germline *Cas9* expression under control of the *vas2* promoter (*Papathanos et al., 2009*) were used as a source of eggs for microinjection or in test crosses described in the text. They were constructed by assembling *vasa* and *Cas9* modules by Golden Gate Cloning into plasmids pDSAY, pDSARN and pDSAP, marked with *3xP3-YFP*, *3xP3-DsRed$_{NLS}$* and puromycin resistance, respectively. The pDSARN-vas2-eSpCas9 plasmid encodes the eSpCas9 variant with reduced off-target activity (*Slaymaker et al., 2016*). Each plasmid was inserted in the X1 *att*P docking site on chromosome 2 as described (*Volohonsky et al., 2015*). The DsRed$_{NLS}$-marked, *vas2-eSpCas9* transgene was then introgressed into the Ngousso background of *A. coluzzii* by 8 successive backcrosses. In the case of the non-fluorescent, puromycin-resistant Cas9 transgene, homozygosity was achieved by first balancing the transgene by crossing to the YFP *vasa-Cas9* line, with puromycin selection of the F1 progeny as described (*Volohonsky et al., 2015*). Counter-selecting YFP fluorescence in the F2 progeny of the self-crossed puromycin resistant F1 yielded the non-fluorescent, vas2-Cas9 homozygous line.

## Mosquito egg microinjection and recovery of transgenics

Embryo microinjection to obtain the *Lp::Sc2A10* transgene by CRISPR-Cas9 knockin was performed as for classical transgenesis (*Volohonsky et al., 2015*) with 400 ng/µl of donor plasmid DNA and addition of 2 µM of the drug Scr7 (APExBio) to the injected plasmid mix. eSpCas9-expressing Ngousso mosquitoes were used as a source of eggs for microinjection (*vas2-eSpCas9, 3xP3-DsRed$_{NLS}$* strain). We injected the knock-in plasmid into 400 eggs, recovered 140 surviving G0 adults and out-crossed them *en masse* to wild-type mosquitoes. In the G1 progeny of the injected G0 males, about 100 GFP positive, putative knock-in individuals were recovered. We established independent families

from ten GFP positive G1 females outcrossed to wild-type, and eliminated the Cas9 transgene (by counter-selecting *3xP3-DsRed*<sub>NLS</sub>) in these families. All 10 founder females were confirmed by PCR to carry the intended transgene insertion in the target *Lp* locus. We pooled two families to yield the *Lp::Sc2A10* line. To avoid confusion between two *DsRed* markers, injection of the *Lp-GD* suppression GD construct (DsRed<sub>NLS</sub>-marked) was performed in eggs expressing wild type Cas9 (vasa-Cas9, 3xP3-YFP G3 background strain). Injection of the *SagGD^zpg* and *SagGD^vasa* constructs encoding their own source of *Cas9* was performed in embryos lacking any Cas9 transgene, but respectively carrying a heterozygous or homozygous copy of *Lp::Sc2A10*. Numbers of injected eggs and recovered transgenics are provided in the main text. Work with genetically modified mosquitoes was evaluated by Haut Conseil des Biotechnologies and authorized by MESRI (agréments d'utilisation d'OGM en milieu confiné #3243 and #3912).

## COPAS analyses

A COPAS Select instrument (Union Biometrica) was used as described (*Marois et al., 2012*; *Bernardini et al., 2014*) to quantify fluorescent transgenes in populations of neonate mosquito larvae or to establish mixed cultures of defined proportions. The use of unfed neonate larvae in clean water was crucial for precise gating and sorting. For sorting, flow rate was kept under 20 larvae/second in Pure mode allowing superdrops. PMTs were set at 500 (GFP), 600–900 (RFP). Delay, Width, signal threshold and minimum TOF parameters were set on 8, 6, 100, and 150, respectively.

## *Plasmodium berghei Pb-PfCSP*<sup>hsp70-GFP</sup> parasite strain

We generated a novel *P. berghei* strain combining the strong GFP fluorescence of *Δp230p-GFP* (*Manzoni et al., 2014*) with the substitution of endogenous *P. berghei* CSP for CSP from *P. falciparum* from strain *Pb-PfCSP* (*Triller et al., 2017*), in which GFP fluorescence was too faint for selecting live infected mosquitoes examined under 488 nm light. For this, 150 mosquito females were allowed to blood feed on a mouse co-infected with both parental parasite strains at a ratio of 1 strongly fluorescent *Δp230p*-GFP for 40 *Pb-PfCSP*. Sexual reproduction of *P. berghei* in the mosquito generates hybrid haploid sporozoites, some of which inherit both *Δp230p*-GFP and *Pb-PfCSP*. Seventen days after their infective blood meal, live female mosquitoes were screened under 488 nm light to select those displaying visible GFP sporozoites trapped at the base of their wing veins. 20 positive females were offered a blood meal on a naïve mouse. When parasitemia reached 0.1%, 3000 strongly GFP positive blood stage parasites were sorted by flow cytometry and injected intravenously into two naïve mice, only one of which developed parasitemia 11 days after passage. Its blood was then passaged into a new mouse. When parasitemia reached 0.4%, 10 new mice were injected with a blood dilution corresponding to 1 parasite each, although this number was probably underestimated as all 10 mice developed parasitemia. Of these, four mice tested PCR positive for *PfCSP* (PCR primers GGCCTTATTCCAGGAATACCAGTGCT / GGATCAGGATTACCATCCGCTGGTTG) and negative for PbCSP (PCR primers GAAGAAGTGTACCATTTTAGTTGTAGCGTC / TGGGTCATTTGGGTTTGGTGGTG). We selected one clone for passage into naive mice, confirmed its PCR negativity for PbCSP and positivity for PfCSP, this clone was called *Pb-PfCSP*<sup>hsp70-GFP</sup>.

## Mosquito infections and bite-back experiments

Neonate larvae from the heterozygous stock of *Lp::Sc2A10* mosquitoes were sorted by COPAS flow cytometry (*Marois et al., 2012*) to assemble a mix of equal numbers of homozygous GFP +larvae (or heterozygous, in some experiments) and negative control siblings. Maintaining the transgene in a heterozygous population ensured the absence of genetic divergence that otherwise may change vector competence between control and transgenic mosquitoes over time. Following COPAS sorting, transgenic and control larvae were co-cultured and adults kept in the same cage to equalize potential environmental influences (including unequal microbial communities) that could differentially affect the vector competence of the two genotypes if grown separately. Mosquitoes were infected together by blood-feeding from the same 12- to 24-week-old CD1 mouse, with a parasitemia between 2 and 4%. Male and female mice were used indiscriminately. Sixteen to 19 days after infection, and 1 day before biting new naive mice, mosquitoes were cold-anesthetized and examined under 488 nm light. *Lp::Sc2A10* females were separated from negative control females based on fluorescence of the transgenesis marker in their eyes. In addition, any mosquito lacking visible GFP parasites (oocsts in

the abdomen, sporozoites visible through the cuticle in wing veins or salivary glands) was discarded to ensure that mice were exposed only to *Plasmodium*-infected mosquitoes. On day 17–20, naive mice were individually exposed to groups of 10 *Lp::Sc2A10* or negative control infected mosquitoes. The number of engorged mosquitoes was recorded after infectious feeding, only mice bitten by at least 6 mosquitoes were included in subsequent analyses with the exception of two mice bitten by 5 and 4 highly infected transgenic and control mosquitoes, respectively, as in this experiment the mouse bitten by the 4 control mosquitoes became infected. Mouse parasitemia was monitored by flow cytometry using an Accuri C6 SORP flow cytometer between day 4 and 12 after bite-back. Mice reaching >1.5% parasitemia were sacrificed, and mice remaining negative 12 days post infectious feeding were considered uninfected. Work on mice was evaluated by the CREMEAS Ethics committee and authorized by Ministère de l'Enseignement Supérieur et de la Recherche (MESRI) under reference APAFIS #20562–2019050313288887 v3.

## Mass spectrometry analysis of hemolymph

Hemolymph from 25 to 30 cold-anesthetized female mosquitoes was collected directly into 1 x Laemmli buffer 48 hr post-blood feeding by clipping the proboscis and gently pressing their abdomen. Samples were precipitated and digested with trypsin, and 1/5 of the digestion product was analyzed on a Q Exactive Plus Mass Spectrometer coupled to an Easy-nanoLC1000 (Thermo). The acquired data was searched against the *Anopheles* UniProt database plus the Sc2A10 protein sequence using Mascot and the total number of spectra corresponding to ApoLpI, ApoLpII, Sc2A10 and a selection of additional hemolymph proteins that served for normalization were counted.

## Protein gels and western blotting

Mosquito salivary glands or carcass were dissected in 20 µl Laemmli buffer, crushed with a pestle and denatured at 65 °C for 5 min. Samples were centrifuged for 3 min at 15,800 *g*, loaded on a Mini-PROTEAN TGX Stain-Free Precast Gel (BioRad) along with 6 µL of PageRuler Plus Prestained Protein Ladder (ThermoFisher) as standard and electrophoresed at 170 V using the Mini Trans-Blot cell system (BioRad). Gels were blotted on PVDF membranes (Trans-Blot Turbo Mini 0.2 µm PVDF Transfer Pack; BioRad) using the mid-range program of a Pierce Fast blotter (Thermo Fisher Scientific). Membranes were blocked for one hour in PBS + 0.1% Tween (PBST) supplemented with 5% fat-free milk powder, incubated overnight at 4 °C with primary antibody, washed three times in PBST and incubated 1 hr at room temperature in secondary antibody conjugated to horseradish peroxidase (HRP). Antibodies were diluted in PBST, 3% milk. Membranes were then washed three times for 10 min with PBS. Antibody binding was revealed using the Super signal WestPico Plus kit (Thermo Fisher). After a 1–2 min incubation, images were acquired using the Chemidoc software (Bio-Rad). Before incubation with further primary antibodies to visualize additional proteins, membranes were stripped for 20–30 min in Restore PLUS Western Blot Stripping Buffer (Thermo Fisher) and washed three times for 10 min in PBST followed by a new blocking incubation. Human antibodies used to control protein loading of salivary gland samples were obtained from 2 ml venous blood taken from a human volunteer arm-feeding *Anopheles* cages over several years. Blood was incubated at room temperature for 1 hr to allow coagulation and centrifuged for 10 min at ~16,000 *g*. Serum supernatant was aliquoted and stored at –20 °C until use. For Coomassie staining of electrophoresed hemolymph samples, hemolymph was collected 42 hr post blood feeding by clipping the proboscis of cold-anesthetized mosquitoes and gently pressing their abdomen, collecting drops of hemolymph into a pipet tip containing protein sample buffer (1 µl of sample buffer per mosquito). Samples were denatured for 2 min at 90 °C and hemolymph from 10 mosquitoes was loaded on each lane of a 4–20% 1.5 mm polyacrylamide gel (Invitrogen). After electrophoresis, gels were rinsed in water and stained in SimplyBlue safeStain solution (Thermo Fisher).

## Amplicon sequencing

A 494 bp region of *Saglin* encompassing the 3 gRNA target sites was amplified from genomic DNA extracted from 150 and 750 COPAS-sorted, DsRed negative neonate larvae with the Blood and Tissue kit (Qiagen) using PCR primers <u>ACACTCTTTCCCTACACGACGCTCTTCCGATCT</u>GCAGAAGCAGCT CGACGC and <u>GACTGGAGTTCAGACGTGTGCTCTTCCGATCT</u>GCAGCTGCCGGAAGTGCT (Illumina adapters underlined) and Phire DNA polymerase (Thermo Fisher Scientific), an annealing temperature

of 68 °C and 32 amplification cycles. Resulting amplicons were gel-purified and sent for sequencing using the Genewiz AmpliconEZ service. Returned paired-end reads (250 nt length each) were assembled to form single contigs using PANDAseq version 2.11 *Masella et al., 2012*; unmerged reads were manually combined pairwise and added to the assembled contigs. Data was analysed using a Docker Desktop Personal version 20.10.16 (https://www.docker.com) -based CRISPResso2 version 2.2.8 *Clement et al., 2019* followed by manual curation.

## PCR genotyping of the *SagGD^vasa* gRNA array

Genomic DNA was extracted from 48 individual pupae taken from populations 1 and 2 G29 and from populations 5 and 7 G14 (corresponding to >32 generations following initial transgenesis). A forward primer (GAAGGCGCTGCAGAAGCAGCTCG) was designed in the Saglin 5' region and a reverse primer (gccctccatgcgcaccttgaa) in the *DsRed* cassette (*Figure 9*). PCR was performed in a total volume of 15 µL using GoTaq (Promega) (94 °C for 5 min then 45 cycles at 95 °C for 1 min, 67 °C for 20 sec, 72 °C for 3.5 min) or Phire polymerase (Thermofisher) (98 °C for 2 min then 45 cycles at 98 °C for 15 sec, 71 °C for 10 sec, 72 °C for 1.3 min), with identical results. PCR products were resolved on 1% agarose gels and the major bands at 3.2 and 1.4 kb (*Figure 9*, A and B type, respectively) were excised from the gel, purified and sent for Sanger sequencing (Eurofins GATC, Germany) using one PCR primer. The following amplicons were sequenced (*Figure 9*): two samples of several pooled large -type A- amplicons from Population 2 (pool of samples #2, 3, 4, 5, 6, 8, 9, 11, 12) and from Population 7 (pool of samples #1, 3, 7, 11, 12), as well as 4 small -type B- amplicons of Population 1 individually (samples #1, 2, 4, 6), and a pool of 4 type B amplicons from Population 7 (pooled #2, 4, 5, 6). Sequence alignments to the theoretical original plasmid used for knock-in were performed with SnapGene software. The unique, 20 bp gRNA protospacer sequences in each of the 6 gRNA coding units allowed to pinpoint which of the gRNAs had been deleted. To genotype mosquitoes for the presence of *Cas9*, a 1661 bp fragment of the *Cas9* gene was PCR amplified with primers CAAGAGCA GACGGCTGGAAA and GGGTGGTCTGGTTCTCTCT.

## Acknowledgements

We thank the laboratory of Chris Janse for providing the parental P. berghei-PfCSP strain from which the Pb-PfCSP^hsp70-GFP strain was derived. We thank Andrew Hammond, Kyrous Kyrou and Andrea Crisanti for the gift of plasmid p17410 to subclone the zpg-Cas9 expression cassette, and Eric Calvo for the gift of the recombinant Saglin protein and polyclonal Saglin antibodies. We thank Lauriane Kuhn, Johana Chicher and Philippe Hammann of the Strasbourg IBMC Proteomics Platform for the mass spectrometry sample preparation and analysis. We thank Muriel Philipps and Claudine Ebel from the Illkirch IGBMC flow cytometry platform for sorting P. berghei-infected red blood cells. We thank Mallory Kastner and Julie Fimeyer for help during the project. This work was supported by Agence Nationale de la Recherche, through research grant #ANR-19-CE35-0007-01 GDaMO to EM, the Laboratoire d'Excellence (LabEx) ParaFrap #ANR-11-LABX-0024 to SB, equipment grant #ANR-11-EQPX-0022 for insectarium operation and by funding from CNRS, Inserm, the University of Strasbourg, and from Contrat Triennal Strasbourg Capitale Européenne 2018–2020. Additional funding was awarded to DK by the DFG as a postdoctoral fellowship (#KL 3251/1-1).

## Additional information

### Funding

| Funder | Grant reference number | Author |
| --- | --- | --- |
| Agence Nationale de la Recherche | ANR-19-CE35-0007-01 | Eric Marois |
| Agence Nationale de la Recherche | ANR-11-LABX-0024 | Stéphanie Blandin |
| Agence Nationale de la Recherche | ANR-11-EQPX-0022 | Eric Marois |

| Funder | Grant reference number | Author |
|---|---|---|
| Deutsche Forschungsgemeinschaft | KL 3251/1-1 | Dennis Klug |

The funders had no role in study design, data collection and interpretation, or the decision to submit the work for publication.

## Author contributions

Emily I Green, Investigation, Methodology, Writing - original draft; Etienne Jaouen, Amandine Gautier, Investigation; Dennis Klug, Investigation, Writing - review and editing; Roenick Proveti Olmo, Data curation, Investigation, Writing - review and editing; Stéphanie Blandin, Formal analysis, Funding acquisition, Project administration, Writing - review and editing; Eric Marois, Conceptualization, Formal analysis, Supervision, Funding acquisition, Investigation, Methodology, Writing - original draft, Project administration, Writing - review and editing

## Author ORCIDs

Dennis Klug http://orcid.org/0000-0002-9108-454X
Roenick Proveti Olmo http://orcid.org/0000-0002-3849-8591
Stéphanie Blandin http://orcid.org/0000-0003-4566-1200
Eric Marois http://orcid.org/0000-0003-4147-3747

## Ethics

Work on mice was evaluated by the CREMEAS Ethics committee and authorized by Ministère de l'Enseignement Supérieur et de la Recherche (MESRI) under reference APAFIS #20562-2019050313288887v3. Work with genetically modified mosquitoes was evaluated by Haut Conseil des Biotechnologies and authorized by MESRI (agréments d'utilisation d'OGM en milieu confiné #3243 and #3912).

## Decision letter and Author response

Decision letter https://doi.org/10.7554/eLife.93142.sa1
Author response https://doi.org/10.7554/eLife.93142.sa2

# Additional files

## Supplementary files

- Supplementary file 1. DNA sequences of constructs used in this study.

- Supplementary file 2. Tracking the evolution dynamics of *Lp::Sc2A10* transgene frequency. A parental cage was assembled containing only heterozygous transgenic mosquitoes (G0, transgene frequency = 50%). Neonate larvae of subsequent generations except G5, 6, 8, 9 (N indicates the number of larvae analysed) were analyzed using COPAS flow cytometry. Gates were drawn on COPAS diagrams around clouds of larvae corresponding to homozygous, heterozygous and negative individuals according to the intensity of GFP fluorescence, and the corresponding percentage of objects in each gate was recorded. Percentages were corrected to exclude objects not corresponding to larvae. Transgene frequency per 100 chromosomes dropped from 50% in G0 to 11.35% in G17, an average loss of 2.3% per generation.

- Supplementary file 3. Fertility tests comparing the number of progeny produced by homozygous *Lp::Sc2A10* female versus WT female mosquitoes. Indicated identical numbers of virgin transgenic and WT females were mixed in cages with WT males. After blood feeding, neonate larvae produced by each cage were analyzed by flow cytometry (COPAS) and the numbers of GFP fluorescent and negative larvae were counted using WinMDI software on COPAS files. Identical fertility of the two categories of females would produce 50% of GFP positive progeny. a, b after the replicate number indicate the first and second egg batch, respectively, from the same mosquitoes. Replicate 1 was composed of smaller mosquitoes, due to higher density larval rearing. All other replicates were performed with mosquitoes of standard size. Replicate 4b was performed with older mosquitoes, with >50% of females already dead.

- Supplementary file 4. Mouse parasitemia after infection by mosquito bites. Mice are grouped by categories according to the genotype of the *Plasmodium*-carrying mosquitoes biting them or by the strain of parasite infecting them. Groups of 10 infected mosquito females (transgenic or their wild-type siblings grown in the same culture), were allowed to bite a mouse. Colored cells indicate

the stage at which infected mice were sacrificed to prevent the appearance of malaria symptoms. n.d=not determined. 'D' in the third column indicates a mouse bitten by transgenic mosquitoes that showed a clearly delayed parasitemia compared to the controls. In two cases (indicated in third column), the mouse was bitten by fewer than 6 WT mosquitoes.

• Supplementary file 5. Temporal dynamics of the *Sag-GD^vasa* and *Lp::Sc2A10* transgenes in 8 mosquito populations. Each strip of panels on successive pages corresponds to an independent caged mosquito population. Each panel is a COPAS analysis snapshot of neonate larvae at the indicated generation, obtained as shown on the first page. Successive panels provide a global view of each transgene's evolution on samples of 1000–4000 neonate larvae at each generation, with fluorescence intensity correlating with transgene copy number (GFP marks the *Lp::Sc2A10* transgene on the Y axis, DsRed marks *Sag-GD^vasa* on the X axis). Up to 9 partially overlapping distinct larval populations corresponding to the 9 possible genotypes (0, 1, or 2 copies of each fluorescence, sketched on page 1 and on selected panels duplicated below the original panel) can be resolved on COPAS diagrams. Dots indicated in population 1 panels correspond to debris (dead larvae, egg shells) and are difficult to distinguish from non-fluorescent larvae unless sorting them for verification under the microscope. Yellow arrow in population 2, G14 indicates the appearance of GFP negative larvae that represent a *Lp* homing refractory mutant (see text). Tracking Populations 1 and 2 started shortly after initial transgenesis, by crossing G2 [*Sag-GD^vasa* / Y; *Lp::Sc2A10 / +*] males to WT females (considered the G0 cross for temporal tracking of genotypes). Population 3 was assembled by mixing 20 transgenics from generation 12 of population 2 with 190 WT. Population 4 was assembled by mixing 60 transgenics from generation 12 of population 2 with 189 WT. In these new G0 mixes reproductive success of transgenics proved higher than that of the WT (due to differences in the quality of the parental mosquitoes), as G1 pupae consisted of 189 GFP transgenics for 358 negatives = 34.5% instead of expected 10% (population 3) or 346 GFP transgenics for 304 negatives = 53.2% instead of expected 24.1% (population 4). In all populations, note the initial rapid convergence of genotypes towards *Lp::Sc2A10* (GFP) homozygosity, the rapid initial disappearance of *Sag-GD^vasa* (DsRed) positives lacking GFP, and the persistence of a fraction of DsRed negative larvae.

• MDAR checklist

### Data availability

All data generated or analysed during this study are included in the manuscript and supplementary files.

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
