## [Editor Report]

This study presents the generation of a two-component gene drive for population modification in Anopheles coluzzii, a major malaria vector in Africa. By testing multiple elegant drive designs, the authors convincingly achieve the spread of antimalarial cargos in caged mosquito populations. Overall, this work represents a significant advance towards a possible application of genetic technologies for malaria control.

---

## [Decision Letter]

[Editors' note: this paper was reviewed by Review Commons.]

---

## [Author Response]

Point-by-point description of the revisionsReviewer #1 (Evidence, reproducibility and clarity (Required)):In this study, the authors made a two-component homing modification gene drive in Anopheles coluzii with a different strategy than usual. The final drive itself targets and disrupts the saglin gene that is nonessential for mosquitoes, but important for the malaria parasite. The drive uses several gRNAs, and some of these target the Lp gene where an anti-malaria antibody is added, fused to the native gene (this native gene is also essential, removing nonfunctional resistance alleles at this locus). In general, the system is promising, though imperfect. Some of the gRNAs self-eliminate due to recombination of repetitive elements, and the fusion of the antimalaria gene had a modest fitness cost. Additionally, the zpg promoter was unable to operate at high efficiency, requiring use of the vasa promoter, which suffers from maternal deposition and somatic expression (the latter of which increased fitness costs at the Lp target). The manuscript has already undergone some useful revisions since its earliest iteration, so additional recommended revisions are fairly modest.Line 43-45: The target doesn't need to be female sterility. It can be almost any haplosufficient but essential target (female sterility works best, so it has gotten the most study, but others have been studied too).

We agree. However, this paragraph focused on previous achievements in malaria mosquitoes, for which suppression gene drives spreading lethality rather than female sterility have not been reported to our knowledge. Even the targeting of *doublesex*, which is a sex determination rather than female fertility gene, results in female sterility (Kyrou et al. 2018). However, we inserted the possibility of female killing by X-shredder GD (Simoni et al., 2020).

Line 69: A quick motivation for studying Anopheles coluzii should be added here (since gambiae is discussed immediately before this).

Thank you for drawing our attention to this point. We modified the sentence to:

“Here, we present the engineering of the Lipophorin (Lp) essential gene in Anopheles coluzzii, a prominent member of the A. gambiae species complex and a major malaria vector in sub-Saharan Africa.”

Introduction section: It might be helpful to break up the introduction into additional paragraphs, rather than just two.

We followed this suggestion and broke up the introduction into 5 paragraphs to make it more breathable.

Introduction last part: The last part of the introduction reads more like an abstract or conclusions section. Perhaps a little less detail would fit better here, so the focus can be on introducing the new drive components and targets

We have followed this suggestion and substantially shortened this last part of the introduction.

Line 207-213: This material could go in the methods section. There are some other examples in the results that could be similarly shortened and rearranged to give a more concise section.

We moved the long description from lines 207-213 to the Methods as suggested, and summarized it simply as:

“Only mosquitoes displaying GFP parasites visible through the cuticle were used to infect mice.”

We emphasize this point because in subsequent experiments using *Saglin* knockout mosquitoes, this enrichment for infected mosquitoes will probably attenuate the Plasmodium-blocking phenotype caused by *Saglin* KO, since mosquitoes lacking *Saglin* tend to be less infected (Klug et al., 2023). Elsewhere in the Results, we still provide detailed descriptions of procedures because we believe they aid understanding and assessing the quality of the experiments.

Line 283-287: I couldn't find the data for this.

Indeed we only summarized the data about the progeny of the [zpg-Cas9; GFP-RFP] line crossed to WT, as we didn’t judge these results worth detailing. Here is our record from one such cross:

GFP-RFP females x WT males 486 (50.7%) GFP+ and 472 (49.3%) GFP- larvae

GFP-RFP males x WT females 1836 (48.9%) GFP+ and 1925 (51.1%) GFP- larvae

This shows no significant gene drive. However in these progenies, a few GFP+ and non-RFP larvae, and a few RFP+ non-GFP larvae were noted by visual examination under the fluorescence microscope, without counting them precisely. Their existence testified to some weak homing activity mediated by *zpg-Cas9* in the *Lp* locus.

We modified the sentence as follows to support our conclusion, and we propose to leave these detailed numbers here in our response, which will be published along with the paper.

“In spite of the presence of the zpg-Cas9 and gRNA-encoding cassettes in the GFP-RFP allele, it was inherited in about 50% of male or female progenies, demonstrating little homing activity of the GFP-RFP locus after crosses to WT, except for the appearance of rare GFP-only or RFP-only progeny larvae, …”

Line 291: Replace "lied" with "was".

Done.

Line 356: Homing in the zygote would be considered very unusual and is thus worthy of more attention. While possible (HDR has been shown for resistance alleles in the zygote/early embryo), this would be quite distinct from the mechanism of every other reliable gene drive that has been reported. Is the flow cytometry result definitely accurate? By this, I mean: could the result be explained by just outliers in the group heterozygous for EGFP, or perhaps some larvae that hatched a little earlier and grew faster? Perhaps larvae get stuck together here on occasion or some other artifact? Was this result confirmed by sequencing individual larvae?

We agree with your skepticism, especially given that the same is not seen in Suppl Figure 2A with a similar genotype setup, i.e., the *vasa* gene drive at the *Lp* locus, or in the G1 of populations 6 or 8 at the *Saglin* locus (Suppl. File 2). Unfortunately, it would take too much time at this point to re-create this line (which has been discarded) to re-examine this issue. Therefore, we acknowledge that another explanation than homing in the zygote may account for this result. Based on our empirical experience COPAS outputs are reliable: such outliers from the heterozygous population are usually not seen, and we always sort neonate larvae a few hours from hatching. Those 6% homozygous-looking larvae may come from a contamination with male pupae when female pupae were manually sorted for the cross to WT males, a human error that we cannot exclude. In this case, the true GFP inheritance would be closer to 79% than to 85%. For these reasons, we must back up from our initial statement as follows:

“The progeny of these triple-transgenic females crossed to WT males showed markedly better homing rates (>79% GFP inheritance).”

And edit the figure legend of Figure 4B to account for the alternative possibility of a contamination with males:

“6% of individuals appeared to be homozygous, revealing either unexpected homing in early embryos due to maternal Cas9 deposition, or accidental contamination of the cross with a few transgenic males.”

Results in general: Why is there no data for crosses with male drive heterozygotes? Even if some targets are X-linked, performance at others is important (or did I miss something and they are all X-linked). I see some description near line 400, but this sort of data is figure-worthy (or at least a table).

For the only example of functioning split gene drive at the Lipophorin locus on chromosome III, we do show homing results from heterozygous GD males in Suppl. Figure 2A (91.2% homing in males inferred from ((40.7+53.1+1.8)-50)χ2). We added this calculation of the homing rates in the figure legend. For full drive constructs in the *Saglin* locus on chromosome X (our final functional design), in addition to the data described in the text near line 400, male data showing “teleguided” homing at the *Lipophorin* locus on chromosome II is shown in Suppl. File 2 (see G2 of population 7, showing close to 100% homing at the *GFP* locus); the same data (less easy to assess) being converted into the G2 point of the graphs in Figure5.

Lines 362-367: What data (figure/table) does this paragraph refer to?

We apologize for the fact that this sentence was misleading. In this population, the genotype frequencies were not tracked at each generation but measured once after 7 generations. We rephrased (now lines 401-403) and now provide the measured values directly in the text:

“We maintained one mosquito population of Lp::Sc2A10 combined with SagGD^zpg^ (initial allele frequencies: 25% and 33%, respectively) and measured genotype frequencies after 7 generations. This showed an increase in the frequency of both alleles (G7: GFP allelic frequency = 59.2%, phenotypic expression of DsRed in >90% of larvae, n=4282 larvae),”

Lines 405-406: There may be a typo or miscalculation for the DsRed inheritance and homing rate here. Should DsRed inheritance be 90.7%?

Thank you for spotting this. You are right, DsRed inheritance would be 90.7% if the homing rate were 81.4% as we mistakenly wrote. Actually DsRed inheritance was really 80.7% so our mistake was in calculating the homing rate: 61.4% is the correct value ((80.7-50)χ2), now corrected in the manuscript.

Figure 5: The horizontal axis font size for population 8 is a little smaller than the others.

True. Corrected.

Line 454: In addition to drive conversion only occurring in females and the somatic fitness costs, embryo resistance from the vasa promoter would prevent the daughters of drive females from doing drive conversion. This means that drive conversion would mostly just happen with alleles that alternate between males and females.

We agree with this idea, although the impact of this phenomenon will depend on the extent of resistance allele formation in early embryos. We observed (Figure 6) that failed homing mutagenesis in Saglin is not that intense, the sequenced non-drive alleles that were exposed 1-4 times to mutagenic activity in females either being mostly wild-type, or carrying mutations that often still left one or two gRNA target sites intact and vulnerable to another round of Cas9 activity. Therefore, alleles passed on from female to female may still undergo drive conversion to a large extent, that future experiments may be able to quantify.

Line 481: Deletions between gRNAs certainly happen, but I wouldn't necessarily expect this to be the "expectation". In our 2018 PNAS paper, it happened in 1/3 of cases. There were less I think in our Sciences Advances 2020 and G3 2022 paper. All of these were from embryo resistance from maternal Cas9 (likely also the case with your drive due to the vasa promoter). When looking at "germline" resistance alleles, we have recently noticed more large deletions.

We agree that the early embryo with maternally deposited Cas9 is probably the most prominent source of mutations at gRNA target sites. Perhaps naïvely we imagined that it would be easier for cells to repair two closely spaced DNA breaks by eliminating the intervening sequence, rather than stitching each break individually. Given that we sequenced many alleles carrying a single mutation, the lack of larger deletions may be explained by lower rates of Cas9 activity in *Saglin*, with mostly a single break at a time, due to limiting Cas9 amounts and their partial saturation with *Lp* gRNAs, and/or lesser accessibility of the *Saglin* locus compared to *Lipophorin*… We deleted the phrase “*Contrarily to our expectation*”.

Figure 6C: It may be nice to show the wild-type and functional resistance sequence side-by-side.

Done

Lines 642-644: This isn't necessarily the case. At saglin, the nonfunctional resistance alleles may still be able to outcompete the drive allele in the long run. This wasn't tested, but it's likely that the drive allele has at least some small fitness costs.

We agree. We inserted this comment in a parenthesis in the text (now lines 644-645):

“Unlike the first approach, this design may allow Cas9 and gRNA-coding genes to persist indefinitely within the invaded mosquito population (unless nonfunctional resistance alleles outcompete the drive allele in the long run).”

A few comments on references to some of my studies:Champer, Liu, et al. 2018a and 2018b citations are the same paper.

Duplicate in our reference library. Corrected.

For Champer, Kim, et al. 2021 in Molecular Ecology, there was a recent follow-up study in eLife that shows the problem is even worse in a mosquito-specific model (possibly of interest as an alternate or supporting citation): https://elifesciences.org/articles/79121

Citation added (line 68).

One of my other previous studies was not cited, but is quite relevant to the manuscript: https://www.science.org/doi/10.1126/sciadv.aaz0525This paper demonstrates multiplexed gRNAs and also models them, showing their advantages and disadvantages in terms of drive performance. Additionally, it models and discusses the strategy of targeting vector genes that are essential for disease spread but not the vectors themselves (the "gene disruption drive"), showing that this can be a favorable strategy if gene knockout has the desired effect (nonfunctional resistance alleles contribute to drive success).

Your 2020 study will indeed now be useful to inform the design of multiplex gRNAs for various gene drives designs, in terms of number of gRNAs, distribution of their target sites, necessity to generate loss-of-function rather than functional resistance allele in the target gene (such as our *Lp* and *Saglin* pro-parasitic genes). The notion of Cas9 saturation with increasing gRNA numbers is also important. When we initiated this project in 2018, we only had intuitive notions that multiplex gRNAs could improve the durability of GD and increase the chances of resistance alleles to be loss-of-function. We thus arbitrarily maximized the number of gRNAs for each of the two targets: 3 for each target in one design, 3 and 4 in another, which, according to your modelling, is luckily close to the optimal numbers for each locus. We now cite your paper as a GD design tool in the discussion about pathways to optimizing our system:

“To further optimize GD design, modeling studies can now aid in determining the optimal number of gRNAs in a multiplex, depending on the specific GD design and purpose (Champer et al., 2020).

In addition to this and to the stabilization of multiplex gRNA arrays, other paths to improvement (…)”

This one is less relevant, but is still a "standard" homing modification rescue type drive that could be mentioned (and owes its success to multiplexing): https://www.pnas.org/doi/abs/10.1073/pnas.2004373117The recoded rescue method was also used in mosquitoes (albeit without gRNA multiplexing) by others, so this may be a better one to mention: https://www.nature.com/articles/s41467-020-19426-0

We added the two references on what is now Line 663:

“Lp::Sc2A10 depends on SagGD for its long-term persistence and spread in a population, and SagGD depends on Lp::Sc2A10 as a rescue allele of the essential Lp target for its survival. This design can be seen as a two-locus variation of rescue-type GDs (Adolfi et al., 2020; Champer et al., 2020)”

Referees cross-commentingOther comments look good. One thing that I forgot to mention: for the 7-gRNA construct with tRNAs, the authors mentioned that it was harder to track, but it sounds like they obtained some data for it that showed similar performance. Even if this one is not featured, perhaps they can still report the data in the supplement?

This GD required examination of the mosquitoes at late developmental stages, such as the pupa, to score red fluorescence under control of the *OpIE2* promoter, that is unfortunately late-active when expressed from the *Lp* locus. We precisely scored only the first 128 pupae arising from the progeny of the first obtained G1 [SagGD/+ ; Lp-2A10/+] females crossed to WT males. Among these:

– 115 were GFP+, DsRed+ (89.8%)

– 12 were GFP+, DsRed- (9.3%)

– 1 was GFP-, DsRed- (<1%)

This allowed us to roughly estimate the homing rates at 98.2% at the *Lipophorin* locus and 79.7% at the *Saglin* locus, which is similar to the other construct without tRNA spacers.

These approximate rates were confirmed by visual examination of progenies in two subsequent generations of [SagGD/+ ; Lp-2A10/+] males and females backcrossed to WT.

Reviewer #1 (Significance (Required)):Overall, this study represents a useful advance. Aside from being the first report for gene drive in A. coluzii, it also is the first that investigates the gene disruption strategy and is the first report of gRNA multiplexing in Anopheles. The study can thus be considered high impact. There are also other aspects of the study that are of high interest to gene drive researchers in particular (several drives were tested with some variations).

We are grateful for your positive, constructive and in-depth analysis of our study!

Reviewer #2 (Evidence, reproducibility and clarity (Required)):The authors initially created a transgenic mosquito colony expressing the Sc2A10 antibody fused to the lipid transporter Lipophorin, and tested the transmission-blocking activity of this transgene. Building off of previous findings that the Sc2A10 antibody inhibits sporozoite infectivity when expressed in mosquito salivary glands, the authors showed that found it was also efficient at inhibiting sporozoite infectivity when secreted into the hemolymph expressed under the lipophorin endogenous promoter in An. coluzzii. They then designed and tested two different gene drives utilizing the Sc2A10-Lipophorin fusion protein. In the first, the authors used a recoded allele of Lp-Sc2A10 while simultaneously utilizing gRNAs that targeted endogenous Lp in an effort to select for mosquitoes that expressed transgenic Lp-Sc2A10 due to the essential nature of Lp. However, this drive was unsuccessful because recoded Lp is necessarily heterozygous while the GD is entering the population, and Lp proved to be largely haploinsufficient. Further, the zpg promoter expressing cas9 was not effective in promoting homing of the gRNAs. In the second gene drive that was tested, authors made use of the endogenous Saglin locus, which expresses a natural agonist for Plasmodium, and is thus desirable to target for disruption in a gene drive that aims to reduce vector competence for Plasmodium. This gene drive also uses recoded Lp-Sc2A10 to replace the wild-type Lp allele, thus selecting for Sc2A10 expression, however this drive is not dependent on fitness of individuals with only one functional copy of Lp.The authors discovered that the efficacy of the zpg promoter to drive homing of cas9 is locus-dependent, limiting the success of their gene drive designs. They do show, however, that the Saglin gene drive succeeds at reaching high frequencies in mosquito populations using instead the vasa promoter to express cas9, and that these transgenic mosquitoes are able to reduce infectivity of sporozoites in a bite-back mouse model. However, they observe gene drive refractory mutations in the Lp gene, despite its highly conserved nature, showcasing the difficulty of avoiding drive resistance even in small populations of mosquitoes, and also observed deletions of gRNAs targeting both Lp and Saglin, further highlighting possible shortcomings in gene drive approaches. Together, these findings are useful to the field in walking the readers through an interesting and promising approach for a novel gene drive, and illustrating the challenges in engineering an efficacious and long-lasting drive.Major comments:As the authors are able to observe Plasmodium within mosquitoes, it would be useful to have these data in the manuscript pertaining to the prevalence and intensity of infection in mosquitoes prior to bite-back assays. If there are data or images that the authors could include, it would be helpful to show if there is a possibility that infection intensity is a variable that contributes to whether or not mice develop an infection. It would also be interesting to note whether there is a different in infection (oocysts or sporozoites) between transgenic mosquitoes and wild type mosquitoes.

This is a valuable suggestion. Please note that, in order to evaluate the transmission-blocking properties of the *Lp-2A10* allele (acting at the sporozoite level), we discarded non-infected mosquitoes prior to bite-back experiments, so that infection prevalence was 100% in the mosquitoes retained for the bite-back. We have not systematically compared parasite loads between transgenic and control mosquitoes. In some experiments comparing Lp-2A10 mosquitoes and their control, we dissected a subset of the mosquito midguts after bite-back to visually ascertain that they showed roughly equivalent oocyst numbers between transgenic and controls. However, we have not precisely recorded these data. It is possible that slightly decreased lipid availability in *Lp::2A10* mosquitoes (their *lipophorin* allele producing slightly less Lp than the WT) negatively affects the parasite, as suggested by previous studies highlighting the role of host lipophorin-derived lipids for parasite development in the mosquito (Costa et al., *Nat Commun* 2018; Werling et al. Cell 2019; Kelsey et al. PLoS Path 2023).

In the case of Lp-2A10 mosquitoes additionally containing a GD in *Saglin*, it is expected that they should carry lower parasite numbers than their controls, an effect of the *Saglin* knockout mutation alone (Klug et al., PLoS Path 2023). Re-inforcing the transmission blocking effect of the 2A10 antibody by reducing parasite loads via the *Saglin* KO was indeed our intention. Hence, having selected the most infected mosquitoes for our bite-back experiments likely attenuated this desired effect, but we still observed a 90% transmission decrease when the two modifications were combined, compared to a 70% decrease with Lp-2A10 alone. We do not plan to perform additional infections experiments for the current manuscript on *Plasmodium berghei* expressing Pf-CSP, but we do intend to record parasite counts in a follow-up study with an optimized *SagGD* transgene and *Plasmodium falciparum* infections. This will be of high relevance for potential future applications in malaria control.

The authors also go into significant detail in the discussion exploring ideas of how to optimize or improve this specific gene drive design. The authors should also stress further the applicability of their discoveries in other gene drive designs, and emphasize the lessons they learned in the difficulties encountered in this study and how these findings could guide others in their decision making process when choosing targets or elements to include in a potential gene drive approach.

We feel that we already emphasized these lessons in the manuscript, in the discussion and when justifying the chosen strategies in the Results section. Lessons for future designs include:

- inserting an antimalarial factor into an essential endogenous gene, preserving its function, can provide many benefits (high expression level, secretion signal that can be hijacked, endogenous introns can be hijacked to host a marker, inactivation by mutagenesis or epigenetic silencing being more difficult…);

- a distant-locus gene drive (as here in *Saglin*) could potentially drive several antimalarial cargoes at the same time, inserted in different loci;

- non-essential mosquito genes agonistic to *Plasmodium* are attractive host loci for a GD, an already old idea illustrated here by the case of *Saglin;*

- multiplex gRNAs are a viable approach to reduce the formation of GD-resistant alleles in essential genes and/or to increase the frequency of loss-of-function alleles, which will either disappear if the gene is essential or decrease vector competence if the gene is pro-parasitic. Hence gRNAs targeting intron sequences should be avoided in order to preserve this benefit, as illustrated by one of our *Lp* gRNAs targeting the first intron and that contributed to generate the only *Lp* viable resistance allele identified in this study;

- To increase long-term stability of the GD construct, repeats should be minimized in gRNA multiplexes through the use of a single promoter and various spacers (tRNAs, ribozymes?) – it remains to be seen if the 76-nucleotide gRNA constant sequence itself, necessarily repeated, will stimulate unit losses in a gRNA multiplex;

- The best promoter to restrict Cas9 expression to the germ line may be *zpg* in some but not all loci; the *vasa* promoter causing maternal Cas9 deposition may still be envisaged if resistance allele formation can be prevented by other means (targeting hyper-conserved essential sequence, multiplexing the gRNAs against an essential gene…).

Minor comments:Line 44 – female sterility but also female killing approaches to crash pop. like X shredder, if authors would like to expand

Female killing citation of Simoni et al., 2020 added (line 45).

Lines 48-60 – Authors should add some references from the literature surrounding ethics and ecology studies related to gene drive release

we added: (e.g., National Academies of Science, Engineering, and Medicine, 2016; Courtier-Orgogozo et al., 2017; de Graeff et al., 2021) on lines 49-51.

Line 114 – Given the only moderate impacts of Saglin's role in Plasmodium invasion, I am not sure this saglin deletion is a convincing benefit for GD as it is probably not impactful enough alone – can the authors soften this statement?

While it’s correct that *Saglin* KO mosquitoes show a significant decrease only in *P. berghei* oocyst counts and not in prevalence when mosquitoes are heavily infected, they do show a significant decrease in both counts and prevalence upon infection with *P. berghei* and, most importantly, *P. falciparum* when parasite loads are lower —a situation that is more physiological (e.g. prevalence of 65% and 13% in WT and Sag(-)KI mosquitoes, respectively, upon infection with *P. falciparum* – Klug et al., PLoS Path 2023). Therefore, for human-relevant *P. falciparum* infections, an impactful decrease in vector competence can be legitimately expected.

Line 126 -Can the authors provide rationale for expressing Sc2A10 with Lp instead of expressing it from salivary glands?

There are three reasons for this. First, we knew from the cited Isaacs et al. papers that the 2A10 antibody was efficient against transmission when expressed in the fat body, and from unpublished work (Maria Pissarev, Elena Levashina and Eric Marois) that anti-CSP ScFvs expressed in the fat body of transgenic mosquitoes blocked sporozoite transmission as efficiently as when expressed from salivary glands. This is certainly favored by the easy sporozoite accessibility to the antibody when both are in mosquito hemolymph. Of note, the transmission blocking results suggest that the binding of ScFv to CSP withstands the crossing of the salivary gland epithelium by sporozoites. Second, we were looking for a host gene expressed as high as possible to produce high levels of Sc2A10 antibody. Third, the host gene must be essential so that resistance alleles would not be viable.

We agree that it would also be possible to use a salivary gene instead of *Lp* as a host for this antimalarial factor. In this case, a same-locus gene drive may have functioned, but the advantages of the host locus being an essential gene would be lost, at least partially, as genetic ablation of the salivary gland, albeit slowing blood uptake, does not prevent mosquito viability and reproduction (Yamamoto et al., PLoS Path 2016).

Line 140 – Can authors give any comment on why these regions of Lp were chosen to be recoded / targeted with gRNAs?

inserting Sc2A10 just after the cleaved Lp secretion signal, and N-terminally to the rest of the Lp protein, was the goal, so that 2A10 would be secreted together with Lp and separated from both signal peptide and Lp by naturally occurring proteolysis. This constrained the choice of the target site to be at the junction between signal peptide and the remainder of Lp protein. An alternative design could have been to insert it between the two subunits ApoLpI and ApoLpII, with duplication of the protease cleavage site, or on the C-terminal extremity of the protein, but there would have been no intron in the immediate vicinity to knock-in a selection marker at the same time.

Line 171 – "stoichiometric"

Corrected.

Line 186 – Can the authors comment or speculate on why the expression levels of the fusion protein are expected to be lower than endogenous Lp?

We did not expect this. It is hard to predict whether and explain how insertion of exogenous sequences in a gene can alter its expression. Possible explanations include: the existence of harder-to-translate mRNA sequences in the Sc2A10 moiety; the addition of seven exogenous amino acids on the N-terminal side of ApoLpII (mentioned in M&M) possibly modifying the stability of the Lp protein; the modification of the intron sequence perturbing efficient intron excision and/or pre-mRNA expression due to the disruption of regulatory elements or to the new presence of the GFP gene in the antisense orientation (albeit expressed in the nervous system and not in the fat body); the presence of the exogenous *Tub56D* transcription terminator used to arrest *GFP* transcription possibly possessing bidirectional termination activity and lowering the mRNA level of the *Lp* allele…

Line 211 – Why were 6 mosquitoes used for these assays, and 10 mosquitoes used in later assays (Line 223)?

Mice were always exposed to groups of 10 mosquitoes, but not all 10 mosquitoes were necessarily biting the mice. We retained mice bitten by at least 6 mosquitoes for further analysis (M&M, lines 871-873 of the revised file).

Line 212 – I would also suggest using letters (Suppl. Table 2A,B,C etc) to refer the specific experiments and sections in the Table.

Implemented.

Line 225- 228 – The authors should mention in the text that homozygotes and heterozygotes do not differ in infection assays.

Added: “Therefore, heterozygous mosquitoes showed a transmission blocking activity comparable to that seen in homozygotes.”

Line 249 – Can the author comment on the impacts of population influx / exchange on the idea that the GD cassette need only be transiently in the population?

If *Lp::Sc2A10* is fixed in the population and the GD gone, indeed an influx of WT alleles through mosquito immigration will begin to replace the antimalarial factor and drive it to extinction due to its fitness cost. As mentioned in the final paragraph of the discussion, this could be seen as an advantage to restore the original natural state—hopefully after malaria eradication! However, we regard a situation where *Lp::2A10* never reaches fixation as more likely, with its spread being re-ignitable by updated GDs (line 741 of the revised file).

Line 273 – Can the authors comment on why this may have occurred more frequently than the expected integration of the GD cassette?

When a chromosome break is repaired, each side of the cut must recombine with the repair template. A possible explanation for our observation is that one side of the break recombined with the injected repair plasmid, while the other recombined with the intact sister chromosome (physiologically probably the preferred option). Since this situation still leaves truncated chromosomes, another repair event can join the plasmid-bearing chromosome end to the sister chromosome. The observation that complex rearrangement occurred frequently suggests that such events can be very common, but will usually go undetected due to the absence of genetic markers. Here, GFP on the intact sister chromosome served as a genetic marker to betray its unexpected involvement in the repair process.

Line 314 – Not all fitness costs are apparent through standard laboratory rearing as was performed in Klug et al. Authors could consider "no known fitness cost" instead.

We agree. This is what we meant by “no fitness cost in laboratory mosquitoes”. We changed this to “no fitness cost at least in laboratory conditions (Klug et al., 2023)” to make clear that this was tested.

Line 407 – don't start new paragraph (same with 409)

We removed these two lines, as we realized they contained an error, and made a correction on line 420 of the revised manuscript.

Line 408 – I'm not sure it's clear why all these populations were kept for a different number of generations – can the authors clarify?

Populations 1 and 2 were the oldest founder populations, therefore maintained for the longest time. As described in the text, all other populations were derived from populations 1 and 2 later in time by outcrossing a subset of individuals to WT mosquitoes. For these derived populations, we reset the clock of generation counting to 0 as we monitored the homing phenomenon “from scratch” in transgenic males crossed to WT, and in transgenic females crossed to WT. Resetting the clock resulted in an apparent lower number of generations for these derived populations. In addition, some of them were discarded early, usually after reaching a stable state, as it was difficult to maintain so many populations in parallel over a long period of time.

Line 558 – "10/12 mice" not immediately clear – the authors could be more specific about how data was combined here

Thank you for pointing out this ambiguity. We replaced by: “the absence of infection in a total of 10 out of 12 mice showed…” (line 561)

Line 586 – Since there do appear to be some fitness costs associated with the Sc2A10 version of Lp, might it be expected that fitness costs imposed by the transgene itself could lead to selection pressures leading to its loss? Or do the authors think that these fitness costs are prevented from causing selection against Sc2A10 due to the design of the transgene such that its translation is a prerequisite for Lp's translation? Is the fact that its removal occurs more rapidly than Lp's any indication that selection against the persistence of Sc2A10 may occur?

Yes, we believe that *Lp::Sc2A10* will progressively disappear, replaced by the WT allele, as shown in Figure 1C, in the absence of a GD stimulating its maintenance and spread. In the *Lp::Sc2A10* transgene, translation of Sc2A10 is indeed a prerequisite for Lp translation, imposing a degree of genetic stability of this transgene in terms of sequence integrity, but this does not mean that the locus cannot be outcompeted by the WT under natural selection, so that long-term persistence of *Lp::Sc2A10* depends on the presence of the GD, as outlined in lines 669-672. As the GD itself can disappear due to the accumulation of resistance alleles, we expect a progressive lift of its pressure to maintain *Lp::Sc2A10* and both loci to be progressively lost, a form of reversibility that may be regarded as desirable (lines 773-776 in v2, 741-743 in v3). Alternatively, both transmission blocking alleles could be maintained by releasing an updated version of the dual GD.

Line 659 – add some further detail to this – how do you envision this to occur?

We have deleted this paragraph, as it hypothesized that *SagGD* could frequently be transmitted to the next generation in the absence of *Lp::2A10*, which is not the case (it would be lethal, and *Lp::2A10* homing is anyway extremely efficient). After a putative field release of [*SagGD* / Y; *Lp::2A10/ Lp::2A10*] males, both transgenes should rapidly be introgressed in the field’s genetic background.

Line 635 – Long paragraph, should be broken up or removal of text. Some of these ideas could possibly be made more concise to improve readability. There are many different hypotheticals that are expanded upon in the discussion.

We admit that this paragraph in the discussion was long and dense. We have split it into 4 smaller paragraphs to better separate the concepts that we want to discuss, and have deleted the part mentioned in the above point.

Line 677 – This scenario seems potentially unrealistic considering the only subtle impacts of Saglin deletion on vector competence, and the potential for population exchange in mosquito populations to dilute out these alleles if the drive begins to fail. Can the author comment or potentially decrease emphasis on such scenarios?

While *Saglin* KO mosquitoes show a moderate decrease of infection prevalence in the context of high infections, the *Saglin* KO decreases parasite loads in all cases, and most importantly, also prevalence upon physiological infections with *P. falciparum* (Klug et al., PLoS Path 2023 and see our response to your comment to line 114 above). This yields a higher proportion of non-infected mosquitoes. Therefore, the impact of *Saglin* mutations should be stronger for the epidemiology of human infections with *P. falciparum* than in laboratory models of infections where parasite loads are very high.

We agree that mosquito migration in natural populations would progressively dilute out the beneficial alleles once the GD effect ceases. The epidemiological impact is difficult to predict and will strongly depend on the durability of the GD and on the intensity of genetic influx from adjacent mosquito populations.

Line 708 – Can the authors speculate on why zpg is sensitive to local chromatin and elaborate on possible solutions or consequences for other drive ideas? This seems broadly important.

We do not precisely know why the *zpg* promoter is more sensitive to local influences than the *vasa* promoter, but this phenomenon seems common for other promoters as well (e.g., the *sds3* promoter as opposed to the *shu* promoter in *Aedes aegypti* (Anderson et al., Nat Comm 2023)). It is possible that the vasa promoter is better insulated from local repressive influences, perhaps by insulating elements akin to *gypsy* insulators in *Drosophila*. Knowledge of genetic insulators active for mosquito genes is lacking as far as we know. Characterization of efficient mosquito insulators, for example if one could be identified within *vasa*, and their combination with *zpg* or *sds3* promoter elements, could potentially improve the locus-independent activity of such promoters. Alternatively, a natural and ideal promoter may still be found showing both an optimal window of expression of Cas9 in the germline, and little susceptibility to local repression.

Line 737 – The suggestion of releasing laboratory-selected resistance alleles in the absence of further context may be provocative and unnecessary here.

We didn’t intend to sound provocative, but are interested in the idea of simple resistance alleles with limited sequence alteration that could be selected in the lab, and released to block a gene drive that turned undesirable, so we wanted to share it with the reader. Mutations in the *Lp* and *Saglin* loci, preserving their functions, can be limited to one or few nucleotide changes in the gRNA target sites, as illustrated by the mutants we sequenced. Lab population of GD mosquitoes can, therefore, be a source of GD refractory mutants that could be leveraged in recall strategies.

Line 850 – unnecessary comma

Corrected.

Line 854 – change to "after infection, moquitoes were "

Changed.

Figure 1 – Not clear what is intended to be communicated by shapes portraying proteins / subunits – consider more detailed illustration of mosquito fat body cells synthesizing and secreting proteins rather than words in text box with arrow to clearly demonstrate the point of this figure.

We propose a new version of figure 1 to better illustrate the fat body origin of Lp and 2A10. We have also re-worked the graphic design to improve several figures.

Figure 3 – I recommend rearranging this figure so that B comes before C, visually. The proportions for the design of in B should also match those used for A.

We have followed these recommendations in the new Figure 3, and also used more logical color codes for the gRNAs and their target genes.

Figure 5 – It is unclear to me why some Populations were maintained for such different lengths of time.

Same point as above for line #408: Populations 1 and 2 are the oldest founder populations, therefore maintained for the longest time. As described in the text, all other populations were derived from populations 1 and 2 later in time by outcrossing to WT mosquitoes, resulting in a lower number of generations for these derived populations. In addition, some of them were discarded earlier, usually after reaching a stable state, as it was not possible to maintain so many populations in parallel for a long period of time.

Figure 7 – Ladder should be labeled on the gel. It may also be helpful for the author to indicate clearly exactly which mosquitoes were shown by sequencing to have these different deletions, as it is occasionally unclear based on band sizing.

We have added the ladder sizes as well as a numbering of individual mosquitoes on Figure 7. We sequenced 4 gel-purified small -type B- amplicons of Population 1 individually (#1, 2, 4, 6), and a pool of 4 type B amplicons from Population 7 (pooled #2, 4, 5, 6) as well as two samples of several pooled gel-purified large -type A- amplicons from Population 2 (pool of samples #2, 3, 4, 5, 6, 8, 9, 11, 12) and from Population 7 ( pool of #1, 3, 7, 11, 12). This information now also appears in the material and methods section (PCR genotyping of the *SagGD^vasa^* gRNA array).

Line 996 – given that there is a size band on the right line of this gel also, can authors crop the gel image to eliminate unnecessary lanes a and b from this figure without losing information needed to interpret this blot?

We agree that this would make the message easier to understand, but cropping lanes a and b would place WT control and Lp::Sc2A10 homozygotes on two separate images, even if a size marker is present on each. We prefer keeping the raw image to facilitate direct comparison of the band sizes, making clear that this was a single protein gel.

Line 1070 – 12 out of how many sequenced mosquitoes?

12 mosquitoes from each of these four populations served as PCR templates to generate figure 7. A subset of amplicons were sequenced individually or pooled, as described above and now in Methods. All sequencing reactions of type A and type B amplicons showed consistent results.

Line 1078 – Can remove some detail like % of agarose, and replication of results with different polymerase as these are already in methods.

Done.

Line 1098 – "Unbless"

Corrected

Reviewer #2 (Significance (Required)): This study illustrates a wide range of issues pertinent for gene drive implementation for malaria control, and as such is of value to the field of entomologists, genetic engineers, parasitologists and public health professionals. The gene drive designs explored for this study are interesting largely from a basic biology perspective pertinent mostly to specialists in the field of genetic engineering and vector biology, but highlight challenges associated with this technology that could also be of interest to a broader audience. A transmission blocking gene drive has not yet been achieved in malaria mosquitoes, and is thus a novel space for exploration. As a medical entomologist that works predominantly outside of the genetic engineering space, I have appreciated the detail the authors have provided with regard to their rationale and findings, even when these findings were inconsistent with the authors' primary objectives or expectations.

Thank you for your positive assessment and for this in-depth evaluation of our data.

Reviewer #3 (Evidence, reproducibility and clarity (Required)):The study by Green et al. generated a gene drive targeting both Saglin and Lipophorin in the Anopheles mosquito, with a view to blocking Plasmodium parasite transmission. This is a highly complex but elegant study, which could significantly contribute to the design of novel strategies to spread antimalarial transgenes in mosquitoes.Overall, this is a complex study which, for a non-specialist reader gets quite technical and heavy in most parts. Despite this, there are key points showing that suppression gene drive may not be the way forward in this instance. However, I would advise explaining certain elements in more detail for the benefit of the general readers. I only have minor points for the authors to address:1) Please point out for the general reader that Anopheles coluzzii belongs to the gambiae complex, since you explain that gambiae are the major malaria spreaders in sub-Saharan Africa.

Done in the introduction (lines 71-73) also in response to Rev. 1

2) The authors pretty much give all results in the last part of the introduction, could the intro be shortened by removing these parts, or just highlighting in a single paragraph the main take home message?

We have condensed this part to highlight the take home messages in the last paragraph, also in response to Rev. 1.

3) Why is Vg mentioned? It is only mentioned once and doesn't have any other mention through the manuscript.

This introduces the two proteins that are by far the most abundant, and present at similar levels, in the hemolymph of blood-fed females, Vg being also prominent on the Coomassie stained gel of Figure 1. We mention Vg also because it represents another excellent candidate locus to host anti-plasmodium factors, as discussed later on lines 600-610 of the Discussion section.

4) Please make it clearer for non-specialists why Cecropin wasn't used.

On lines 630-636 we explain that we decided to leave out Cecropin to avoid potential additional fitness costs due to expression at all life stages in the fat body, as opposed to solely in the midgut after blood meal (Isaacs et al. PNAS 2012); and to avoid complexifying the anti-Plasmodium *Lipophorin* locus in a way that could further reduce the functionality of the *Lp* gene. We also had prior knowledge from unplublished work that Sc2A10 alone was sufficient to block sporozoite infectivity.

5) Why were homozygous and not heterozygous transgenics transfected if there is such as fitness cost to homozygous mosquitoes?

The fitness cost of homozygous mosquitoes is actually mild, unnoticeable if homozygotes are bred in the absence of competing heterozygotes and wild-types (lines 151-156). Microinjection experiments to obtain the different versions of *SagGD* were, therefore, performed on either the heterozygous or homozygous line. As for infection assays, the anticipated effect of gene drive is to promote homozygosity at the Lp::Sc2A10 locus. For this reason, it made sense to test the vector competence of homozygotes, in addition to the fact that the Plasmodium-blocking phenotype was expected to be stronger (and thus, easier to document) with two copies of the transgene. Only after obtaining a large dataset from infection assays with homozygotes did we test heterozygotes and found that they actually had a similar phenotype.

6) Line 211 – what was the average number of infected mosquitoes used per infection for each mosquito strain?

As described in the text (lines 204-206 of v2; 208-212 of the revision) and in the Methods (lines 868-873), non-infected mosquitoes were discarded prior to performing the experiment using 10 infected mosquitoes per mouse, and we discarded mice bitten by fewer than 6 mosquitoes. So at least 6 infected mosquitoes bit each mouse (often 8-9).

7) Line 219 – please be clearer regarding this being infection detected in the blood.

We replaced « infection » with « detectable parasitemia in the blood »

8) Line 320 – please clarify why the zpg promoter was used.

The advantages of *zpg* are mentioned in lines 257-258 and 320-322 (revised file).

9) Line 375 – what was the rationale for using so many gRNAs?

3 or 4 gRNAs against *Lipophorin* and 3 gRNAs against *Saglin*, amounting to a total of 6 or 7 gRNAs against the two loci. The rationale is explained on lines 249-253 : the goal was to maximize the chance of causing loss-of-function mutations in the essential *Lp* gene and to favor elimination of GD resistant alleles by natural selection, in case of failed homing. For *Saglin* which is a non-essential gene, we wanted to ensure loss-of-function of failed homing alleles to achieve a reduction in vector competence, even if GD-resistant alleles accumulate. We sought to make this rationale clearer by adding a sentence on lines 328-332:

“Multiplexing the gRNAs was intended to promote the formation of loss-of-function alleles in case of failed homing at the Lp and Saglin loci: non-functional alleles of the essential Lp gene would be eliminated by natural selection while non-functional Saglin alleles would reduce vector competence.”

Line 555 – please state how long post bite back parasite appears in infected mice.

We changed this sentence to “…two of these six mice developed parasitemia six days after infection” (line 556).

Reviewer #3 (Significance (Required)):This is potentially a highly significant study that could provide a vital mechanism for generating efficient gene drives. Although highly technical and complex in most parts, with a little clarification in certain areas this manuscript could be of great value to a general readership.

Thank you for your appreciation and thoughtful evaluation of our manuscript.

Reviewer #4 (Evidence, reproducibility and clarity (Required)):The authors hijacked the Anopheles coluzzii Lipophorin gene to express the antibody 2A10, which binds sporozoites of the malaria parasite *Plasmodium falciparum*. The resulting transgenic mosquitoes showed a reduced ability to transmit Plasmodium.The authors also designed and tested several CRISPR-based gene drives. One targets Saglin gene and simultaneously cleaves the wild-type Lipophorin gene, aiming to replace the wildtype version with the Sc2A10 alele while bringing together the Saglin gene drive.Drive-resistant alleles were present in population-caged experiments, the Saglin-based gene drive reached high levels in caged mosquito populations though, and simultaneously promoted the spread of the antimalarial Lp::Sc2A10 allele.This work contributes to the design of novel strategies to spread antimalarial transgenes in mosquitoes. It also displays issues related to using multiplexing gene-drive designs due to DNA rearrangements that could prevent the efficient spread of the gene drive in the long term.This is tremendous work considering how many transgenic lines and genetic crosses are performed using mosquitoes. The conclusions are supported by the data presented, and some modifications regarding the experimental design description through text/figure improvements would facilitate the reading and flow of the paper.Here some questions/comments:Line 124-125: Reference?

Added

Line 133-134: Reference?

Added

Table 1: It seems the authors have some issues recovering a good amount Sc2A10 from hemolymph samples. Is this a problem of the antibody per se? Is it the Lp endogenous promoter weak? Could this be improved by placing the antibody in a different genomic region? Alternatives could be discussed.

The 2A10 antibody must be initially produced in the same, very high, amounts as the Lp endogenous protein with which it is co-translated. Therefore, its low relative abundance must result from faster turnover or stickiness to tissue, as hypothesised on lines 176-177. We believe that virtually any other endogenous promoter would be weaker than *Lp* and produce lower Sc2A10 levels.

Figure 1B: It would be nice to have a representation of the genome after integration. You could add a B' panel or just another schematic under the current one.

In agreement with this suggestion and that of rev. 3, we added a new panel in 1B.

Supplementary Figure 1b: Could the authors explain the origin of the (first) zpg promoter used? Is it from An. Coluzzii? It seems they use a different one in the gene drive designs later (see comments below too).

Yrou et al., from genomic DNA from our colony of *A. coluzzii*. The resulting promoter fragment harbored several single nucleotide polymorphisms (SNPs) compared to published sequences, as typically observed when cloning genomic fragments due to high genetic diversity in Anopheles species. Such SNPs are not usually expected to affect promoter activity, but are difficult to distinguish from PCR mutations which, in turn, could decrease or abolish promoter activity if mutating an essential transcription factor binding site. For this reason, our next constructs were based on the validated zpg sequences from Kyrou et al. The first cloning strategy was described in the Results section but was missing in the material and method section. This is now corrected (lines 773-779).Figure 3: Please, correct to A, B, C order. Current one is A, C, B.

Done.

Could the authors include a schematic of the final mosquito genome after integration? I can see they are targeting two different locations (Saglin and Lp). It is unclear though from the figure where the Sc2A10-GFP is coming from. I understand this represents the mosquito genome as you injected heterozygous animals already containing the Sc2A10-GFP. Maybe label the Sc2A10-GFP as mosquito genome or similar? A schematic showing mosquito embryos already carrying this and then the plasmid being injected could help.

Figure 3 does not represent the injection of new transgenic constructs. Instead, it shows the conversion process of chromosomes X and II in a germ cell carrying both transgenes in the heterozygous state, to illustrate how the dual gene drive can spread in a population after WT mosquitoes mated with transgenics carrying both the SagGD and Lp-2A10 alleles. We have re-worked the graphic design of this figure and modified its title to make this more clear.

Line 330-331: Do you know the transgenesis efficiency? Did the authors make single or pools for crossing and posterior screening? It would be interesting to know about transgenesis rates to inform the community.

We no longer perform single crosses for transgenesis, as batch crosses ensure higher recovery of transgenics due to the collective reproductive behavior (swarming) in *Anopheles*. Therefore, we cannot precisely calculate the transgenesis efficiency. However, >60 positive G1s from a pool of 36 G0 males crossed to WT females is indicative of a rather high integration efficiency. We consistently observe high efficiency of transgene integration when using the CRISPR/Cas9 system, that we estimate to be about 5-fold more efficient than docking site transgenesis, and much more efficient than *piggyBac* mediated transgenesis.

Line 357/Figure 4B: Could the authors explain in the text GFP+ vs. GFP++?

GFP++ was meant to indicate higher intensity of GFP fluorescence than GFP+, due to two copies of the transgene versus one, but see our response to reviewer 1’s comment to line 356 about the questionability of homing in the zygote.

Line 357: Where is the vasa promoter that made the "rescue" coming from? Is it amplified from Coluzzii? Please, include this explanation for clarification. Why the authors think the zpg from Kyrou et al. 2018 works for the cassette integration but not for homing? They discuss positional effects, any references showing that?

We amplified the *vasa* promoter from *A. coluzzii* using primers CggtctcaATCCcgatgtagaacgcgagcaaa and CggtctcaCATAttgtttcctttctttattcaccgg (annealing sequence underlined) to have a fragment equivalent to that (*vas2*) characterized in Papathanos et al., 2009. We have now added this information in the Methods under *Plasmid construction*. This is the only source of vasa promoter used in this work.

About *zpg* promoter activity : we have past experience suggesting that promoters, such as the *hsp70* promoter from *Drosophila*, can be sufficient to express enzymatic activities in embryos injected with helper plasmids, even though the same promoters appear to become inactive once integrated in the genome. This may be due to injected “naked” plasmids being readily accessible to the transcription machinery, unlike organized chromatin. A recent reference showing genomic positional influences on promoter efficiency is Anderson et al., 2023, which we have added on line 710 of the Discussion.

Line 362: No reference to figure nor table.

These data (numbers from a COPAS analysis) are provided directly in the text in this sentence (which has been clarified in response to Reviewer 1). See lines 364-369 of the revision.

Line 417: The text brings the reader back to Figure 3C. Could the authors move this panel for easier flow of the paper?

We agree that positioning of this panel in Figure 3 is a bit awkward, but this western blot pertains to the characterization of the insertion shown in Figure 3. Placing it after COPAS analyses would be equally awkward.

Line 472-474: How many WT alleles were recovered? It is not stated unless I missed anything, which is possible.

We refrained from providing a quantification of this, and focussed on qualitative results, as we didn't trust the quantitative representativity of our high-throughput amplicon sequencing results in terms of allele frequency in the sampled mosquito population. A large fraction of sequenced reads corresponded to PCR artefacts such as primer dimers and unspecific short amplicons, potentially affecting the relative frequencies of gene-specific amplicons. However, among the sequenced gene-specific amplicons, WT alleles were the majority (lines 474-475).

Figure 5. Could the authors discuss why the observed DsRed-gene drive drop in population 1 at ~18 generation? The population gets to the point where only 50% of the population carries the Cas9-DsRed cassette. Considering that the Saglin gene drive only converts through females (inserted into the X chr.), and some indels could be generated by generation 20, how do you explain the great recovery until fully spreading into the population?

We agree that this is somewhat puzzling. We don’t have a satisfactory explanation beyond stochastic effects, possibly promoted by population bottlenecks: although we strived to maintain these populations at a high number of individuals at each generation, we cannot exclude that at a given generation only a relatively small fraction of individuals contributed to the next generation, leading to fluctuations in allelic frequencies. This would be possible particularly for populations 1 and 2, which were not monitored frequently between generations 10 and 18, at which point additional populations 5-8 were established and it was decided that close monitoring of all populations was important.

It seems to me populations 3-8 are new cage experiments by randomly picking mosquitoes from populations 1 and 2 (at a specific generation) and mixing them with WT individuals. Could the authors explain the reasoning for these experiments? I believe populations 3-8 deserves a different figure (main or supplementary) describing how they were seeded. It is confusing having everything together as these experiments were performed differently way and for a different reason compared to populations 1 and 2. Some cage schematics and drawings would help in understanding the protocol strategy for populations 3-8.

This is correct for populations 3 and 4 that indeed originated from randomly picking mosquitoes from populations 1 and 2 at generation 10 and mixing them with WT individuals. Populations 5, 6, 7 and 8 are crosses between generation 16 transgenic partners of one sex to WT of the other sex, as indicated above the COPAS diagrams provided in Suppl. File 2. We apologize for having insufficiently described how each population was assembled and now provide more details (lines 422-429, in the figure 5 legend, and G0 crosses spelled out on top of each population diagram). In setting up these populations, we wanted to test the effects of various routes by which the transgenes may be introduced into a wild mosquito population: release of unsorted transgenic males + females, or release of one sex only (probably males in the field, but the crosses with transgenic females as with transgenic males also served to re-quantify homing in the second generation of each cross).

The modified text reads as follows:

“Populations 3 and 4 were established by mixing randomly selected transgenic mosquitoes (both males and females of generation 10) from populations 1 and 2, respectively, with wild-types, to mimic what may occur in a mixed-sex field release. Populations 5-8 were established by crossing single-sex transgenic mosquitoes to WT of the opposite sex, both to mimic a single-sex field release and to re-assess homing efficiency after 16 generations.”

Also, could you add homozygous and heterozygous labels in the figure legend to help understanding the different lines.

As indicated on the side of the figure and in the figure legend, lines don’t represent homozygous vs. heterozygous frequency, but allele frequency (continuous lines), and frequency of mosquitoes carrying the transgene (dotted lines). In the figure legend we now provided the calculation formulas for gene frequency: [ 2 x (number of homozygotes) + (number of heterozygotes)] / 2 x (total number of larvae) for the autosomal *Lp::2A10* transgene, and [ 2 x (number of homozygotes) + (number of heterozygotes) ] / 1.5 x (total number of larvae) for the X-linked *SagGD* transgene.

Figure 6: The authors sequenced non-DsRed individuals from generations 3-4. The authors also mentioned they sequenced mosquitoes from generation 32 (Figure 7). Interestingly, they observed that these mosquitoes were missing a piece of the cassette (they contained 2 gRNAs instead of 7). Since the amplicons only cover the gRNA portion, a PCR covering the Zpg-Cas9 portion would be ideal to confirm that only the gRNAs are missing. Sampling DsRed+ mosquitoes from generations 3, 18 and 31 (populations 1 and 2) and carrying out these experiments is recommended. Although unlikely, I would be worried about the Cas9 being deleted due to unexpected DNA rearrangements; in that case, the cassette would contain the DsRed marker alone.

Thank you for this suggestion. We no longer have DNA samples from the earlier generations. Thus, we genotyped 7 DsRed positive male mosquitoes from each of current populations 1, 2 and 7 (generation 41 since transgenesis) for the presence of Cas9. We detected a Cas9-specific amplicon of 1.6 kb in 21/21 sampled DsRed positive mosquitoes, in parallel to the same shortened gRNA arrays detected in earlier generations. This suggests that the Cas9 part of the transgene was not affected by the loss of gRNA units. We made a panel C in Figure 7 showing these results and mentioned them on lines 537-538. Of note, the *Cas9* moiety of the gene drive construct shows no repetitive sequence and should therefore not be as unstable as the gRNA multiplex array. The observed excisions of gRNA expression units were strictly due to recombinations between repeated U6 promoter sequences (Figure 7).

The authors employ 3 different gRNAs that are 43 and 310 nts apart. It has been shown that only 20 nt lack of homology produces an important reduction on gene drive performance (Lopez del Amo et al. 2020, Nat Comms). Also, it has been shown that gRNA multiplexing approaches should be kept with a low number of gRNAs, 2 being maybe the best one depending on the design (Samuel Champer 2020, Sciences Advances). This could be discussed more.

Thank you for this suggestion. These results were not published when this study was initiated, so that our gene drive constructs could only be designed on empirical bases. For gRNA numbers, see the new discussion point and inclusion of a reference to the study by S. Champer *et al.*, on line 700-702. The reduction of drive performance with longer non-homologous stretches is indeed also a very important point, that we now discuss on lines 713-717, citing your study:

“Finally, tighter clustering of gRNA target sites at target homing loci, especially Saglin, should improve gene drive performance by reducing the length of DNA sequences flanking the cut site that bear no homology to the repair template on the sister chromosome and need to be resected by the repair machinery to allow homing (López Del Amo et al., 2020).”

Reviewer #4 (Significance (Required)):There are different novelty aspects from my point of view in this work. While most of the scientists focus on developing CRISPR-based gene drives in An. Stephensi and gambiae, this work employs An. Coluzzii. Some limitations regarding fitness cost associated with the Lp gene were also noted and discussed by the authors.

To be fair, earlier gene drive studies were performed on the G3 laboratory strain, traditionally named *A. gambiae*, although it is probably itself a hybrid strain from *gambiae* and *coluzzii*. Still, the Ngousso strain from Cameroon that was used in this study is thought to be a bona fide *A. coluzzii*. We have also added a reference to a recent paper (Carballar-Lejarazu et al., 2023) that also describes a population modification GD in *A. coluzzii*.

First, they show that An. Coluzzii mosquitoes infect less when containing the antimalarial effector cassette inserted in their genomes. Second, a gene drive is showing super-Mendelian inheritance in An. Coluzzii, which would be the second example of a gene drive in these mosquitoes so far to my knowledge.I believe this is the first manuscript experimentally using multiplexing approaches (multiple gRNAs) in mosquitoes (all previous works I saw were performed in flies). While previous gene-drive works employ only one gRNA in mosquitoes, this works explores the use of different gRNAs targeting nearby locations to potentially improve HDR rates and gene drive spread. Although they observe gene drive activity, they also show DNA rearrangements due to the intrinsic nature of multiplexing gene drives that can generate multiple DNA double-strand breaks, impeding proper HDR and clean replacement of the wildtype alleles. This is important from a technical point of view as it shows this approach requires optimization. They included 3 gRNAs targeting the Saglin gene, and trying 2gRNAs instead could be interesting for future investigations.

We now discussed optimization with the help of modeling, in response to Reviewer 1, on lines 701-702.

This work will be very useful for the CRISPR-based gene drive field, which seeks to develop genome editing tools to control mosquito populations and reduce the impact of vector-borne diseases such as malaria.This reviewer intended to understand the work and provide constructive feedback to the best of my abilities. I apologize in advance if I misunderstood anything.

Thank you for your appreciation, insight, and constructive evaluation of our manuscript.